# Capturing Polysemanticity with PRISM: A Multi-Concept Feature Description Framework

**Laura Kopf**[1,2]     **Nils Feldhus**[1,2]     **Kirill Bykov**[1,2,3,7,8]     **Philine Lou Bommer**[1,3]
**Anna Hedström**[1,3,4,5]     **Marina M.-C. Höhne**[3,6]     **Oliver Eberle**[1,2]

[1]Technische Universität Berlin, Germany     [2]BIFOLD, Germany
[3]UMI Lab, ATB Potsdam, Germany     [4]Fraunhofer Heinrich-Hertz-Institute, Germany
[5]ETH AI Center, Switzerland     [6]Universität Potsdam, Germany
[7]Munich Center for Machine Learning (MCML)     [8]Technische Universität München
{kopf,feldhus,oliver.eberle}@tu-berlin.de
{kbykov,pbommer,ahedstroem,mhoehne}@atb-potsdam.de

## Abstract

Automated interpretability research aims to identify concepts encoded in neural network features to enhance human understanding of model behavior. Within the context of large language models (LLMs) for natural language processing (NLP), current automated neuron-level feature description methods face two key challenges: limited robustness and the assumption that each neuron encodes a single concept (monosemanticity), despite increasing evidence of polysemanticity. This assumption restricts the expressiveness of feature descriptions and limits their ability to capture the full range of behaviors encoded in model internals. To address this, we introduce Polysemantic FeatuRe Identification and Scoring Method (PRISM), a novel framework specifically designed to capture the complexity of features in LLMs. Unlike approaches that assign a single description per neuron, common in many automated interpretability methods in NLP, PRISM produces more nuanced descriptions that account for both monosemantic and polysemantic behavior. We apply PRISM to LLMs and, through extensive benchmarking against existing methods, demonstrate that our approach produces more accurate and faithful feature descriptions, improving both overall description quality (via a description score) and the ability to capture distinct concepts when polysemanticity is present (via a polysemanticity score).

## 1   Introduction

Large Language Models (LLMs) have rapidly become integral to a range of real-world applications, from software development [1] to medical diagnostics [2]. Despite their growing influence, the internal decision-making processes of these models remain largely opaque. A growing number of approaches aim to understand these black-box systems by analyzing their internal structure in human-interpretable ways, such as mechanistic interpretability [3, 4, 5], structured explanations [6, 7], advanced feature attributions [8, 9], and free-text explanations [10, 11].

A central goal of this research is to assign interpretable, functional roles to individual components such as neurons or attention heads [12, 13, 14, 15]. The presence of *polysemanticity*, the tendency of individual features to encode multiple, semantically distinct concepts or patterns [16, 17], complicates the process of explaining model components, as it defies the common assumption that a single neuron is associated with a single function or pattern. From a coding-theoretic perspective, this reflects how neural capacity is distributed across multiple tasks. While several concept extraction techniques like sparse autoencoders (SAEs) [18, 19] aim to disentangle polysemantic features, many learned features still encode multiple concepts [20], and therefore cannot be regarded as truly monosemantic.

39th Conference on Neural Information Processing Systems (NeurIPS 2025).

**Extracting Feature Descriptions**

Layer 47, Feature 3815
{GPT-2 XL, MLP}

PRISM

"Quantities, specifically numbers, and time periods"

"Personal experiences or opinions"

"Indefinite articles before contextual nouns"

Multi-concept feature descriptions

**Evaluation**

Polysemanticity Scoring

"quantities"

"experiences"

"articles"

$\theta$

Lower Cosine Similarity → high polysemanticity

Description Scoring

Control dataset
"quantities"
"experiences"
"articles"

Higher activation → more accurate description

Figure 1: Overview of the PRISM framework. PRISM captures multiple concepts per feature, enabling the detection of both polysemantic and monosemantic features, unlike prior approaches that constrain each feature to a single description. For example, feature 3815 in layer 47 was previously labeled as monosemantic [20], whereas PRISM reveals that it responds to multiple distinct concepts. Polysemanticity scoring summarizes how diverse the concepts associated with a feature are, while description scoring assesses how well each concept aligns with the feature's activation distribution.

While the problem of polysemanticity is generally addressed through the extraction of sparse features, such as via sparse autoencoders, feature description methods, which aim to explain the functional purpose of individual features, typically provide a single explanation per feature [21, 22, 23, 24, 25]. This can limit the ability to capture the full range of patterns a feature may represent. To offer more comprehensive and nuanced feature descriptions, we introduce PRISM, a framework for generating and evaluating multi-concept feature descriptions that considers *multiple* patterns per feature. In Figure 1, we present a neuron previously labeled as "possibly monosemantic" [20] that PRISM reveals to activate for a highly diverse and heterogeneous set of concepts characterized by textual descriptions. Throughout this work, a concept refers to a cluster of token-level inputs that elicit similar contextualized feature activations, allowing each feature to be associated with multiple concepts that reflect the diversity of inputs it responds to. Our contributions include:

(1) A framework, PRISM, that generates multi-concept descriptions of features (Section 3), providing greater precision and granularity compared to existing methods.

(2) A set of quantitative evaluations (Section 3.2) for comparing multi-concept textual descriptions across different feature description methods.

(3) A first multi-concept feature description analysis of language model features, revealing that individual features encode a highly diverse and heterogeneous set of concepts (Section 5).

By addressing the limitations of single-concept feature descriptions, PRISM enables a systematic and nuanced understanding of internal representations, crucial to advance the interpretability of language models. Our code is made publicly available to the community.[1]

## 2 Related Work

**Automated Interpretability**  Recent generative AI systems have grown increasingly complex, motivating efforts to scale interpretability. In addition to methods that provide local explanations for individual predictions, a complementary line of research focuses on fully automated descriptions of model components like neurons and their associated functions. In this context, an automated interpretability approach was introduced to generate a textual description for each neuron in GPT-2 XL [21]. This method has since become a foundation for subsequent work on feature descriptions [26, 19, 23, 22, 27, 25].

**Feature Description Evaluation**  Automated feature description methods often rely on simulation-based approaches, where a model predicts neuron activations for given inputs [21]. This approach, widely adopted in recent work [26, 19, 23], evaluates descriptions based on the correlation between simulated and actual activations. Other evaluations include binary classification of activating and non-activating contexts [28], contrastive methods using distractor samples [29, 30], and combined approaches that assess input capture accuracy and feature characterization precision [25, 31, 32]. Simulation-based approaches herein rely on an external model's ability to accurately predict neuron

---

[1]https://github.com/lkopf/prism

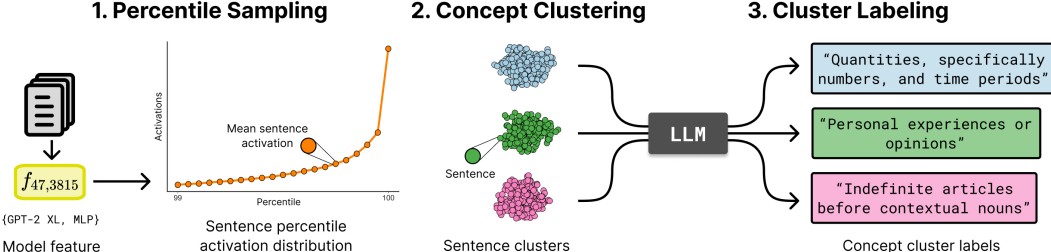

Figure 2: Steps for extracting feature descriptions with PRISM. In Step 1, PRISM processes a text dataset through the model and selects sentences from the top percentile of the activation distribution for a given feature. In Step 2, these high-activation sentences are embedded using a sentence encoder and clustered to identify recurring patterns. In Step 3, the top activating examples from each cluster are used to prompt an LLM, which generates descriptive labels for each cluster.

activations given a feature extracted by an additional explainer model. Currently, it remains unclear how well LLMs reliably predict activations of other models, adding uncertainty and complexity to the evaluation of feature descriptions. Our approach instead directly compares feature activation distributions to controls, using both parametric and non-parametric evaluation measures.

**Clustering and Topic Analysis in Text**    Recent work in topic modeling has relied on unsupervised clustering of language model embeddings to discover latent topics without relying directly on keywords [33, 34]. While methods like LLooM [35] and goal-driven explainable clustering [36] use LLMs to extract or assign high-level textual summaries, our work focuses on fine-grained interpretability of internal model features.

For an extended discussion of related work, see Appendix A.1; additional details on feature description methods are provided in Table 2 in Appendix A.2.

## 3   PRISM: A Framework for Multi-Concept Feature Descriptions

We introduce PRISM, a framework for generating multi-concept descriptions of model features (Section 3.1), and evaluating their quality through polysemanticity and description scoring metrics (Section 3.2).

**Preliminaries**    Let a decoder-only LLM be defined as function $f : \mathcal{X} \to \mathcal{Z}_1 \times \cdots \times \mathcal{Z}_L$, where $\mathcal{X}$ is the input space (i.e., token sequences), $L$ is the total number of blocks, and $\mathcal{Z}_\ell \subseteq \mathbb{R}^{d_\ell}$ denotes the hidden representation at layer $\ell \in \{1, \ldots, L\}$. We denote the layer-specific subfunction as $f_\ell : \mathcal{X} \to \mathcal{Z}_\ell$, such that the output of layer $\ell$ is $z_\ell = f_\ell(\mathbf{x})$. We define a *feature* as the activation of a single neuron $i \in \{1, \ldots, d_\ell\}$, corresponding to the scalar function $f_{\ell,i} : \mathcal{X} \to \mathbb{R}$, defined as $f_{\ell,i}(\mathbf{x}) = z_{\ell,i}$ which denotes the activation of the $i$-th neuron for the $\ell$-th layer.[2] A *feature description method* aims to associate each feature with a single or multiple human-interpretable textual descriptions. Mathematically, each feature $z_{\ell,i}$ is assigned a subset of $1, \ldots, N$ possible descriptions by a set-valued description function $\phi_\lambda : \mathcal{F} \to \mathcal{P}(S)$, such that $s_{i_\ell} = \phi(f_{i_\ell}; \lambda)$, where $\phi$ takes as input the feature function $f_{\ell,i} \in \mathcal{F}$, $\lambda$ represent method-specific hyperparameters[3] and $\mathcal{P}$ is the power set of all valid description subsets $\mathcal{S}$, including empty and all. This formulation accommodates both single- and multi-concept descriptions, depending on the implementation of the description function $\phi$.

### 3.1   Extracting Feature Descriptions

To address feature polysemanticity, we propose a method capable of capturing multiple concepts per feature. Let $f_{\ell,i} \colon \mathcal{X} \to \mathbb{R}$ be the *fixed* feature under consideration, where $\ell \in \{1, \ldots, L\}$ and $i \in \{1, \ldots, d_\ell\}$. Given a corpus $\mathcal{D} \subset \mathcal{X}$, denote the multiset of its activations by

$$A_{\ell,i} = \big\{ f_{\ell,i}(\mathbf{x}) \ \big| \ \mathbf{x} \in \mathcal{D} \big\}. \tag{1}$$

---

[2]In the case of a sparse autoencoder (SAE) feature, let $l : \mathcal{Z}_\ell \to \mathbb{R}^k$ be a learned sparse encoder that maps the $\ell$-th layer's activations to a $K$-dimensional sparse encoding, where the $j$-th SAE feature is defined as $f_{\ell,j}(\mathbf{x}) = l_j(f_\ell(\mathbf{x}))$ where $l_j(\cdot)$ denotes the $j$-th SAE neuron.

[3]We avoid defining this further to avoid notational clutter.

Our multi-concept framework consists of the following steps in reference to Figure 2:

1. **Percentile Sampling.** Let $\mathrm{perc}_q(A_{\ell,i})$ be the operator, returning the $q$-th percentile[4] of the empirical distribution of $A_{\ell,i}$. Fix a grid of high-percentile levels

$$\mathcal{Q} = \{q_1, \ldots, q_m\} \subset [0, 1], \qquad q_1 < \cdots < q_m. \tag{2}$$

   The *high-activation sample set* for the neuron is

$$\mathcal{T}_{\ell,i} = \left\{\mathrm{perc}_q(A_{\ell,i})\right\}_{q \in \mathcal{Q}} \tag{3}$$

2. **Concept Clustering.** Let $e\colon \mathcal{X} \to \mathbb{R}^{d_e}$ be a sentence-embedding function. For a pre-specified $k \in \mathbb{N}$ we partition the embedded set $\left\{e(\mathbf{x}) \mid \mathbf{x} \in \mathcal{T}_{\ell,i}\right\}$ into clusters $\mathcal{C}_{\ell,i} = \{C_1, \ldots, C_k\}$ by employing K-Means method [38].

3. **Cluster Labeling.** Let $\mathcal{G}$ denote a large language model acting on sets of sentences. For each cluster $C_k$ we select the $N_s$ sentences in $C_k \in \mathcal{C}_{\ell,i}$ with the highest activations and query

$$s_j = \mathcal{G}\left(\mathrm{top}_{N_s}(C_j)\right), \tag{4}$$

   where $s_j \in \mathcal{S}$ is a concise natural language summary of the common theme in $C_j$.

### 3.2 Evaluating Multi-Concept Feature Descriptions

**Polysemanticity Scoring**   To quantify the degree of polysemanticity, we measure the similarity among the generated descriptions per feature. Descriptions are encoded using a sentence embedding model, and their pairwise cosine similarities $\tau_j = e(s_j), \tau_j \in \mathbb{R}^T$ with $i \neq j, i, j \in \{1, \ldots, k\}$ are computed:

$$\cos(\theta) = \frac{\sum_t^T \tau_{i,t} \tau_{j,t}}{\sqrt{\sum_t^T \tau_{i,t}^2} \sqrt{\sum_t^T \tau_{j,t}^2}}, \tag{5}$$

where $T \in \mathbb{N}$ is the dimension of the sentence embedding. Features with lower average similarity scores are considered more polysemantic, while high similarity indicates monosemanticity.

**Description Scoring**   Following previous work, which uses contrastive methods to differentiate between activating samples and control samples [29, 30], we adapt the COSY evaluation method [39] to language models (see Figure 7 in Appendix A.3). COSY evaluates each candidate feature using two complementary metrics. The Area Under the Receiver Operating Characteristic (AUROC) measures how well the feature distinguishes between control data points $\mathbb{X}_0$, consisting of 1,000 randomly sampled entries from Cosmopedia [40], and target concept data points $\mathbb{X}_1$, consisting of 10 concept-specific text samples generated by an LLM for the feature description. Given the corresponding activations $\mathbb{A}_0 \in \mathbb{R}^n$ for $\mathbb{X}_0$ and $\mathbb{A}_1 \in \mathbb{R}^m$ for $\mathbb{X}_1$, the AUROC is computed as

$$\Psi_{\mathrm{AUROC}}(\mathbb{A}_0, \mathbb{A}_1) = \frac{\sum_{a \in \mathbb{A}_0} \sum_{b \in \mathbb{A}_1} \mathbf{1}[a < b]}{|\mathbb{A}_0| \cdot |\mathbb{A}_1|}. \tag{6}$$

The Mean Activation Difference (MAD) quantifies the normalized difference between the mean activation on the target and control datasets:

$$\Psi_{\mathrm{MAD}}(\mathbb{A}_0, \mathbb{A}_1) = \frac{\frac{1}{m} \sum_{b \in \mathbb{A}_1} b - \frac{1}{n} \sum_{a \in \mathbb{A}_0} a}{\sqrt{\frac{1}{n-1} \sum_{a \in \mathbb{A}_0} (a - \bar{a})^2}}, \tag{7}$$

with mean control activation $\bar{a} = \frac{1}{n} \sum_{a \in \mathbb{A}_0} a$. In our evaluation, we report the percentage of features with positive MAD scores, i.e., the fraction of features satisfying $\mathbb{A}_1 > \mathbb{A}_0$. See Appendix A.3 for dataset, model and activation details.

## 4   Quantitative Evaluation

In the following, we quantitatively evaluate our proposed feature description method, PRISM, against existing approaches for neuron and SAE feature interpretation.

---

[4]In practice we estimate $\mathrm{perc}_q$ online via the P$^2$ algorithm [37].

## 4.1 Experimental Setup

In our experiments, we evaluate PRISM against competitive feature description methods, MaxAct[5], GPT-Explain [21], Transluce-Explain [23], Neuronpedia [42], and Output-Centric [25][6]. Unless otherwise specified, all experiments are conducted on the English training subset of the C4 CORPUS [41], a large, cleaned version of Common Crawl's web crawl corpus. We process text excerpts to a uniform length of 512 tokens, either by truncation or padding, ensuring consistency across model inputs. Feature activations are extracted from four pre-trained LLMs, from three layers per model (early, middle, late)[7]. For GPT-2 XL [43] and Llama 3.1 8B Instruct [44], we analyze MLP neurons. For GPT-2 Small [22] and Gemma Scope [45], we focus on residual stream SAE features on account of the publicly available implementations. More details on the evaluation procedure and experimental setup are provided in Appendix A.3 and Appendix A.4.

For the practical implementation of each step in the PRISM framework we choose the following settings:

- For **(1) Percentile Sampling**, we identify all text excerpts whose mean activation values fall within the 99th–100th percentile, sampling one excerpt per percentile bin with a step size of 1e-05, resulting in 1000 high-activation excerpts per feature. Compared to top-$k$ sampling, this approach captures a broader spectrum of strong activations, allowing us to collect meaningful text samples from a wide range of conceptual patterns.

- For **(2) Concept Clustering**, the resulting text set is embedded using the gte-Qwen2-1.5B-instruct sentence transformer [46], and then k-means clustering is applied with $k = 5$ to uncover recurring conceptual patterns. Using a fixed number of clusters strikes a balance between granularity and human interpretability, enabling multiple semantic patterns to emerge while reducing redundancy or incoherence. When clusters are highly similar, the resulting low polysemanticity score reflects monosemanticity.

- For **(3) Cluster Labeling**, to generate *human-interpretable labels*, we prompt a large language model (Gemini 1.5 Pro [47][8]) using the $N_s = 20$ text excerpts with the highest mean activations for each cluster. Only positive activations are considered, and a token is highlighted if its activation exceeds a sample-specific threshold, defined as the 90th percentile of the sample's activation distribution. Token spans corresponding to peak activations are highlighted with square brackets "[...]" [49, 28, 23] in the prompt to guide the model. The LLM is instructed to output a concise summary of the shared concept in each cluster. Additional prompt details are provided in Appendix A.4.

## 4.2 Ablation Studies

To assess the robustness of our framework, we conduct ablation studies along two dimensions: (i) varying the number of clusters used for description generation, and (ii) replacing the language models used in both description generation and evaluation. Full experimental details and results are provided in Appendix A.5

**Cluster Size**   We vary the number of clusters $k$ used in concept clustering to analyze its impact on interpretability (Table 3). Larger $k$ improves best-case description quality by isolating more fine-grained activation patterns, but reduces average interpretability as coherent patterns are split across clusters, revealing a tradeoff between precision and coverage.

**Text Generators**   To evaluate the impact of language model choice on our framework, we ablate the text generators used in both description generation and evaluation. For description generation (Table 4), Qwen3 32B [50] achieves performance comparable to Gemini 1.5 Pro (the default in our original implementation), while Phi-4 [51] and DeepSeek R1 [52] follow similar qualitative trends,

---

[5]For obtaining feature descriptions with the MaxAct method, we use a subset of 10,000 samples from the English training split of the C4 CORPUS [41], collect the top five activating samples per feature, and generate feature descriptions using the same prompt as in PRISM(see Appendix A.4).

[6]We use descriptions generated by their Ensemble Raw (All) method, which is best performing on their input-based evaluation.

[7]More details can be found in GitHub Repository `https://github.com/lkopf/prism`

[8]The model originally employed in this study (Gemini 1.5 Pro) was deprecated upon completion of the project. Consequently, subsequent experiments, including those pertaining to MaxAct and output-centric evaluation, were conducted using the available and comparable model, Gemini 2.0 Flash Lite [48].

Table 1: Benchmarking feature description methods. AUROC reflects classification performance, while MAD measures activation differences between target and control descriptions. PRISM (max) reports the best-matching score per feature; (mean) averages over all descriptions. Values are means across selected features from early, middle, and late layers. AUROC includes 95% confidence intervals; MAD represents the percentage of positive MAD scores. Bold indicates best performance; dashes denote unavailable descriptions for certain models. See Appendix A.4 for details.

| Method | GPT-2 XL (MLP neuron) | | Llama 3.1 8B Instruct (MLP neuron) | | GPT-2 Small (resid. SAE feature) | | Gemma Scope (resid. SAE feature) | |
|---|---|---|---|---|---|---|---|---|
| | AUROC (↑) | MAD (↑) | AUROC (↑) | MAD (↑) | AUROC (↑) | MAD (↑) | AUROC (↑) | MAD (↑) |
| MaxAct | 0.53 (0.49-0.58) | 11.86% | 0.54 (0.46-0.63) | 50.00% | 0.53 (0.49-0.58) | 11.86% | 0.60 (0.50-0.69) | 50.00% |
| GPT-Explain [21] | 0.64 (0.56-0.73) | 65.00% | — | — | — | — | — | — |
| Transluce-Explain [23] | — | — | 0.59 (0.51-0.67) | 63.33% | — | — | — | — |
| Neuronpedia [42] | — | — | — | — | 0.54 (0.50-0.59) | 18.97% | **0.62 (0.53-0.72)** | **63.33%** |
| Output-Centric [25] | — | — | 0.55 (0.46-0.64) | 58.33% | 0.57 (0.53-0.62) | 22.03% | 0.58 (0.49-0.67) | 46.67% |
| PRISM (mean) | 0.65 (0.61-0.69) | 66.33% | 0.52 (0.48-0.55) | 51.33% | 0.51 (0.50-0.53) | 13.22% | 0.43 (0.39-0.46) | 24.67% |
| PRISM (max) | **0.85 (0.78-0.91)** | **91.67%** | **0.71 (0.63-0.78)** | **81.67%** | **0.57 (0.53-0.61)** | **28.81%** | 0.54 (0.45-0.62) | 38.33% |

showing that the approach does not depend on a single model. For evaluation (Table 5), alternative LLMs yield slightly lower absolute scores but preserve the relative ranking of methods. These results confirm the robustness of our evaluation setup and the generalizability of the framework across language models.

### 4.3 Sanity Checks

Prior work has shown that LLMs can produce plausible yet unfaithful explanations unrelated to the prompt [53, 54, 55, 11, 56]. Like most feature description methods, our approach relies on LLM-generated feature descriptions. To address potential issues of faithfulness, we conduct two sanity checks that assess description reliability under controlled settings: (1) randomizing sentences within clusters and (2) randomizing cluster descriptions. We further evaluate the polysemanticity scoring, examine feature description quality across percentile activation intervals, and analyze relative activations. Details on the experimental setup, results, and analysis are provided in Appendix A.6.

Randomizing sentences or descriptions yields AUROC values near random relative to the baseline (Table 6). While MAD scores are often not statistically significant due to large standard deviations, they consistently decrease, aligning with our expectations. For polysemanticity, randomly assigned descriptions yield notably lower similarity values than true scores, reflecting reduced semantic coherence (Figure 13). Examining percentile intervals, the top 25% (0.75–1.0) of activations closely match baseline performance, whereas lower quartiles show reduced scores, demonstrating a positive correlation between activation strength and description quality (Table 7). The analysis of relative activations between the 99th and 100th percentiles shows that extremely small ratios are rare, especially in early layers, indicating that percentile-based sampling reliably captures diverse, meaningful patterns (Figure 14).

### 4.4 Benchmarking Experiment Results

Table 1 compares the performance of PRISM against GPT-Explain and Output-Centric. We report two variants: PRISM (max), which reflects the score of the best-matching description, and PRISM (mean), averaging scores across $k = 5$ descriptions used throughout the paper. All reported values are means over a selected subset of features from three model layers, i.e., early, middle, and late. Dashes (—) indicate that the respective method does not provide feature descriptions for the corresponding model, preventing evaluation.

**Empirical Advantage of PRISM** A key observation is that PRISM (max) achieves the highest AUROC and MAD scores, across all model and feature-type combinations. For example, on GPT-2 XL (neuron features), PRISM (max) reaches an AUROC of 0.85 and a MAD of 91.67%, indicating that its descriptions are more accurate and outperform the competitive approach of GPT-Explain. One exception is observed with Gemma Scope (SAE features), where the Neuronpedia method slightly outperforms PRISM (max), achieving the highest MAD (63.33%) and AUROC (0.62). This suggests that, for certain architectures or feature types, alignment in the output space may offer complementary advantages. Additional analysis on the distribution and variability of AUROC and MAD scores is provided in Appendix A.4. Overall, we find that neuron-based features yield more reliable interpretations than SAE-based features, providing further evidence for recent methodological

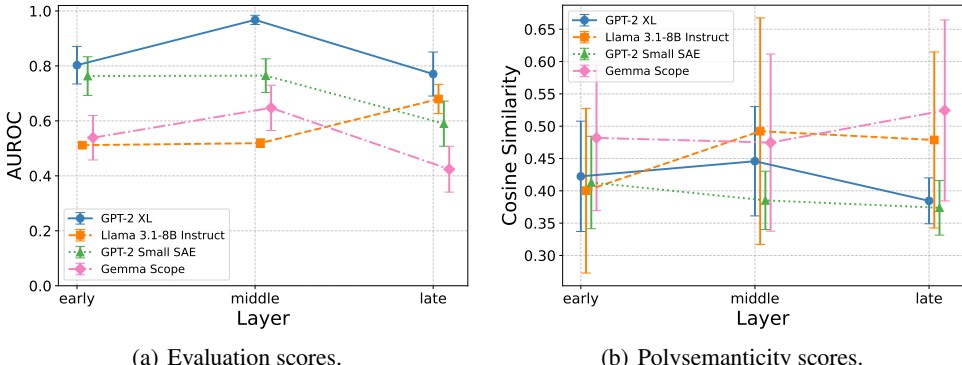

(a) Evaluation scores.
(b) Polysemanticity scores.

Figure 3: Comparison of PRISM (max) AUROC evaluation scores and PRISM polysemanticity scores across different models and layers.

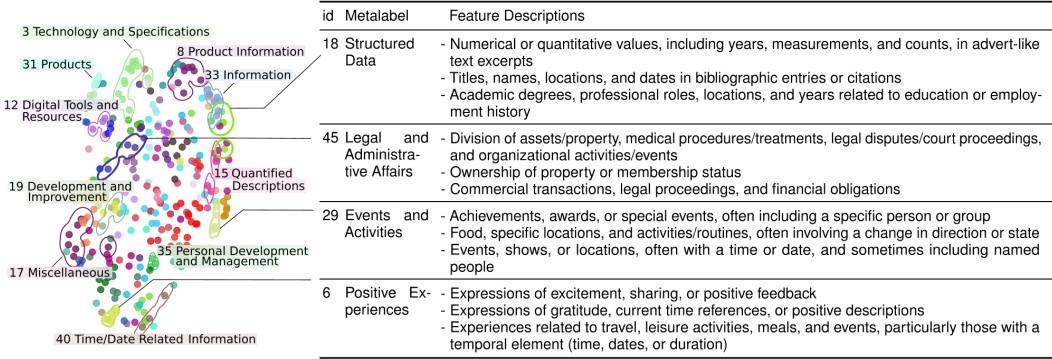

Figure 4: Clustering of identified PRISM feature descriptions in GPT-2 XL. The $k_m = 50$ meta-clusters are visualized using UMAP, with metalabels generated by Gemini 1.5 Pro and three randomly selected sample descriptions shown per cluster.

concerns about achieving improved interpretability through SAEs [57]. Extended results on the output-centric evaluation are provided in Appendix A.4.

**Varying Quality and Polysemanticity across Models** In Figure 3(a) and Figure 3(b), we analyze the feature descriptions' quality and polysemanticity across layers. Judging the evaluation scores, middle layers generally appear to be easier to interpret (Figure 3(a)). In most models, the AUROC scores peak in the middle layer. An exception is observed in Llama 3.1 8B Instruct, where evaluation scores (measured in AUROC, see 3.2) increase in the later layer.

To analyze the degree of polysemanticity, we use the gte-Qwen2-1.5B-instruct sentence transformer [46] to embed the five natural language descriptions generated per feature. We then compute pairwise cosine similarity within each set of five descriptions to estimate semantic consistency. In Figure 3(b), polysemanticity scores show no consistent trend across layers and vary across models and feature types. Interestingly, we can also note that Gemma Scope SAE feature descriptions show the highest monosemanticity across all layers, as shown in Figure 3(b). Despite high variability, our findings suggest that polysemanticity does not consistently increase or decrease with layer depth, and interpretability varies significantly across architectures.

## 5 Investigating Multi-Concept Features

After validating our approach, we can leverage PRISM's novel polysemantic analysis to investigate the diversity of feature descriptions. We also take an initial step toward human evaluation of multi-concept descriptions via polysemanticity judgments.

### 5.1 Exploring Concept Spaces

The distinction between syntax and semantics has deep roots in logic [58, 59] and the study of language [60, 61], where representations are understood to involve both structural and meaning-

bearing properties. Additionally, pragmatics [62] has emerged as another level of analysis, focusing on how context, intention, or social norms shape the interpretation of language. In the context of LLMs, representations have most commonly been categorized as syntactic, including dependency structures and part-of-speech tags [63], or semantic, involving token associations [64], task-specific information [65], or abstract encodings such as goals [66], with recent work also exploring pragmatic, context-dependent interpretations [67].

To gain a better overview of the representational diversity and variety of identified concepts, we now explore all extracted feature descriptions with PRISM. We embed them using GPT-2 XL, extract the final token as a feature description representation, and apply k-means clustering with $k_m = 50$ to identify and visualize meta-clusters using UMAP [68], illustrating the space of learned concepts within the model [69, 70]. In Figure 4, we present a subset of the resulting clusters and their generated metalabels for GPT-2 XL. Further examples can be found in Appendix A.7, including results for GPT-2 Small SAE in Figure 16. Metalabels summarizing shared themes among grouped feature descriptions are generated using Gemini 1.5 Pro, with clusters outlined based on Gaussian kernel density estimation (bandwidth $h = 0.3$) with point weights that decay exponentially with distance from the cluster centroid.

First of all, we find that automatically clustering and summarizing feature descriptions is effective in producing human-interpretable abstractions and associated metalabels, enabling users to reduce the potentially large number of descriptions they need to inspect. When inspecting the resulting clusters, we observe a range of distinct categories. For GPT-2 XL, we observe diverse *semantic* categories, e.g., referring to "Positive Experiences" (id 6) or actions in "Events and Activities" (id 29). Furthermore, we identify domain-specific clusters of feature descriptions, with associated neurons responding to categories such as "Digital Tools and Resources" (id 12) and "Legal and Administrative Affairs" (id 45). Examples of *syntactic* diversity include "Time/Date Related Information" (id 40) and "Structured Data" (id 18), focusing on part-of-speech and specific sentence structure. We further identified combined representations, e.g., syntactic concepts of quantity in the context of specific domains such as food or recipes ("Product Information", id 8). Similar patterns are observed for GPT-2 Small SAE (see Figure 16 in Appendix A.7).

Overall, clustering feature descriptions uncovers multiple levels of analysis, offering a valuable approach for categorizing feature descriptions. Building towards a shared vocabulary to characterize model representations could further reveal universal concepts across models, aligning with recent discussions and evidence on universal representations [71, 72].

## 5.2 Polysemanticity Analysis and Human Interpretation

Following prior work on human annotation of feature descriptions [32, 23], we selected 8 instances as representative samples for obtaining feature label descriptions. Each instance included 5 text clusters generated using Steps 1-3 of the PRISM framework (Figure 2) in the same format the LLM receives during cluster labeling. Seven participants annotated clusters of highly activating sentences and rated the polysemanticity of the resulting labels on an 11-point scale (0.0-1.0), enabling comparison with the PRISM polysemanticity score (Section 3.2). Full study details and results are in Appendix A.8.

Figure 5 compares human-annotated labels with descriptions generated by PRISM. The first example (top row), a polysemantic neuron in GPT-2 XL, receives low scores from both humans (0.40) and PRISM (0.38), reflecting high polysemanticity. The cluster descriptions here are diverse and capture distinct concepts. In contrast, the second example (bottom row) shows a monosemantic SAE feature in Gemma Scope: both human and PRISM scores are high, and all labels consistently reference "time", differing only in context. Additionally, our comparative analysis of polysemanticity scores assigned by humans and models, shown in Figure 6, indicates an overall strong and consistent alignment between human annotations and PRISM scores.

## 6 Discussion and Conclusion

In this work, we propose PRISM, a novel framework for identifying multi-concept feature descriptions in LLMs. Our framework also allows us to ground resulting descriptions in systematic evaluation approaches, by including a description score that provides a description quality measure. Thus, PRISM not only provides an automated interpretability of model components via human-interpretable descriptions, but is the first framework that addresses the challenge of detecting more

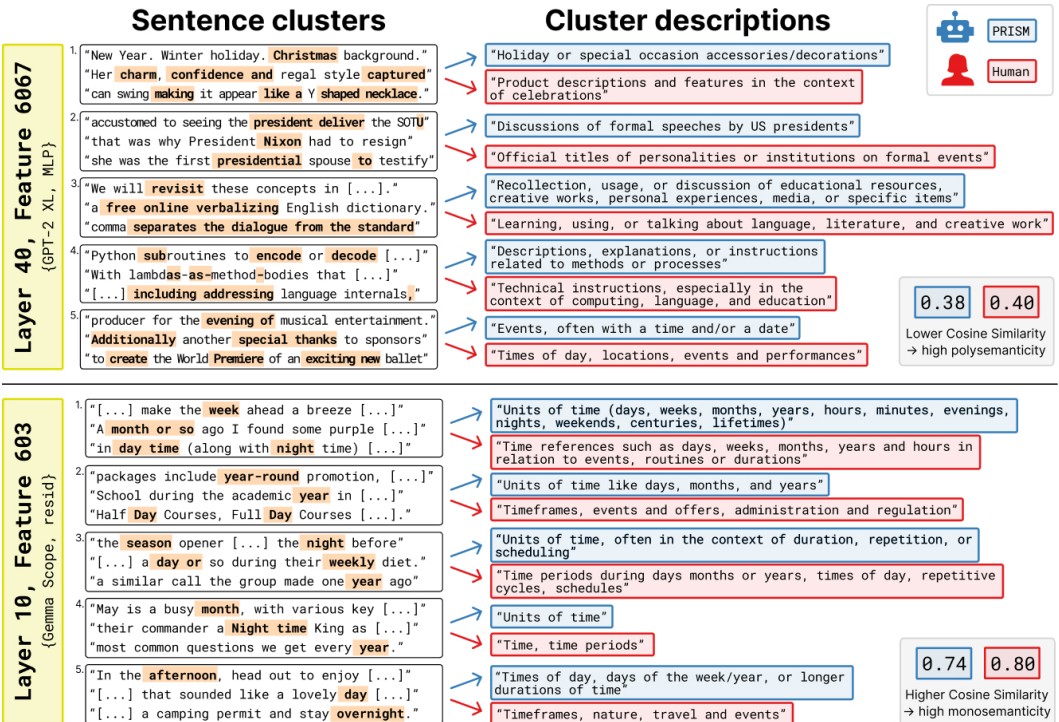

Figure 5: Cluster labeling comparison between human and PRISM (LLM). We compare cluster labeling for two features: a polysemantic feature (top row) and a monosemantic feature (bottom row). On the left, we show representative text spans from input samples that strongly activate the feature, grouped into five clusters based on shared patterns. Within each span, tokens with the highest activations are highlighted. On the right, we compare the cluster labels generated by PRISM (LLM-based) and a human annotator shown the same input as the model. Additionally, the human rates the conceptual coherence of the five cluster labels on a scale from 0.0 to 1.0, where lower values indicate more diverse (polysemantic) and higher values more consistent (monosemantic) labeling. This rating is directly compared with PRISM's polysemanticity score for the same feature.

than one activation pattern a model feature is sensitive to and allows us to quantify variations in the activation patterns using the framework's polysemanticity score.

We conduct several experiments demonstrating the performance and analytical applications of PRISM. Our results show that multi-concept descriptions more accurately distinguish target concepts from control data, with statistically more distinct mean activations. PRISM also extends prior work by capturing variation in description quality [39] and differences in polysemanticity across model layers [73]. Beyond qualitative analysis, PRISM's multi-concept approach and novel polysematicity scoring provide new directions for studying the structure and interpretability of model features. In exploring the concept space, we use PRISM to characterize more complex components, finding and interpreting patterns that specific attention heads or groups of neurons respond to. Deep learning systems utilize distributed features that, in principle, allow for compositionality. By providing a description of more complex sets of interacting components, we can tackle a timely challenge to the community. Lastly, we took a step towards testing the alignment of our PRISM framework with human interpretation. Especially rating polysemanticity is a complex task for humans, even when presented with sentence examples. Our results highlight that the PRISM framework not only provides multiple human interpretable descriptions for neurons but also aligns with the human interpretation of polysemanticity.

**Limitations** As we have focused on textual explanations, our descriptions are limited to concepts expressible in natural language, and thus may be unable to capture complex syntactic structures like graphs or algorithmic concepts where the underlying operation is not easily described in the constrained vocabulary space. Fixing the number of clusters in PRISM cannot guarantee that provided descriptions capture all or the most salient concepts for a feature; instead, they represent a

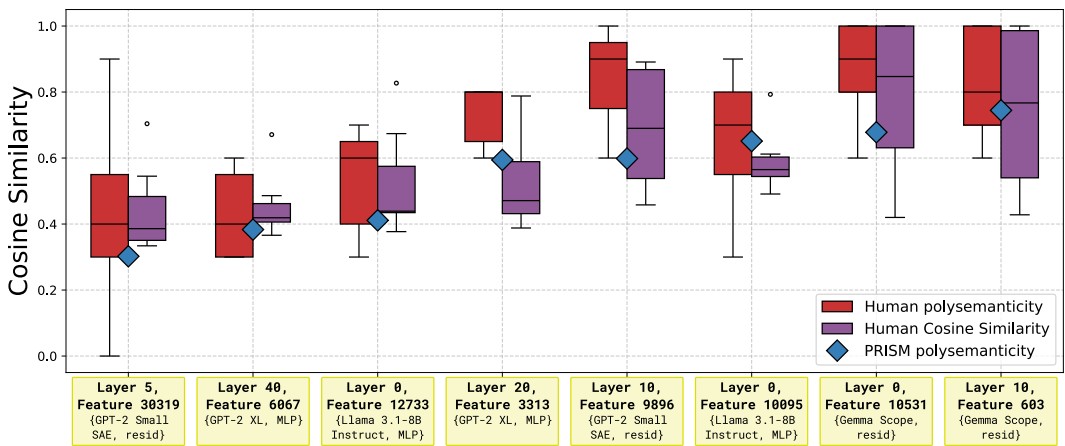

Figure 6: Results of the human evaluation study. Each group on the x-axis corresponds to one feature (layer, feature, model, feature type). For each feature, we show Human polysemanticity scores (red boxplots) from participants' ratings of cluster descriptions, Human Cosine Similarity (purple boxplots) computed with sentence embeddings of the same human-written descriptions, and the corresponding PRISM polysemanticity score (blue point marker). Lower values indicate higher polysemanticity. The results illustrate that features judged by PRISM as polysemantic receive both lower human similarity ratings and lower embedding-based similarity, while features with high PRISM scores show higher human scores.

subset of key concepts the feature is responsive to. Parametric measures such as MAD are sensitive to outliers, especially in NLP settings where activations often follow heavy-tailed distributions. To address this, we pair MAD with the more robust AUROC score, yielding a more comprehensive view of feature behavior. Finally, reliance on maximally activating corpus examples can restrict interpretations to observed concepts, limiting coverage of rare patterns or out-of-distribution effects across datasets [74].

**Future Work** A promising next step is to study how the method's performance changes as the number of clusters varies, and to examine whether alternative clustering techniques can uncover a hierarchical organization of the resulting descriptions. Beyond our quantitative evaluation, we also stress the need for rigorously designed human-evaluation protocols and community benchmarks that provide transparent, standardized measures of progress in automated interpretability.

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

# A Appendix

## A.1 Related Work

A wide variety of works seek to examine the mechanisms of generative AI systems. Many initially focused on attributions in input space [8, 9], with later approaches focused on discovering circuit and graph structures [3, 4, 5], as well as self-explanations generated by the model [10, 11]. To extract relevant representations and model features, a variety of works have explored interpretability for analyzing concepts [75], representations [76], and clustering [77] in the context of attributions of sentence summaries [78] and concept discovery in LLMs [79]. In the following, we will focus specifically on studies and methods centered on feature descriptions.

## A.2 Feature Description Methods

One of the earliest works on feature description in language models is SASC (Summarize and Score) [80], which generates natural language descriptions of neurons in a pre-trained BERT model. Shortly thereafter, an automated interpretability method for describing all neurons in GPT-2 XL was proposed [21]. This approach analyzes the textual patterns that cause a neuron to activate, and uses GPT-4 as *explainer model* to generate a description of the neuron's function. Given a set of token-activation pairs derived from text excerpts and corresponding neuron activations, the *explainer model* identifies common patterns, based on which it generates a textual description of the neuron's role. This method has since been widely adopted and further developed, forming the basis for many subsequent methods targeting both individual neurons [23, 25] and SAE features [26, 19, 42, 22, 28, 27, 45, 29, 25]. An overview of representative feature description methods is provided in Table 2.

Table 2: Representative feature description methods for language models, listed in chronological order, including the model and feature types they target.

| Method | Target Model | Feature Type |
|---|---|---|
| SASC [80] | BERT | neuron |
| GPT-Explain [21] | GPT-2 XL | neuron |
| Pythia SAE [26] | Pythia 70M and Pythia 410M | SAE feature |
| Anthropic SAE [19] | one-layer transformer | SAE feature |
| Neuronpedia [42] | DeepSeek R1 Dist Llama 8B, Gemma 2 2B, Gemma 2 2B IT, Gemma 2 9B, Gemma 2 9B IT, GPT OSS 20B, GPT-2 Small, Llama 3.1 8B, Llama 3.1 8B Instruct, Pythia 70M Deduped, Qwen 2.5 7B IT, Qwen 3 4B | SAE feature |
| GPT-2 SAE [22] | GPT-2 Small | SAE feature |
| GPT-4 SAE [22] | GPT-4 | SAE feature |
| EleutherAI SAE [28] | Llama 3.1 7B & Gemma 2 9B | SAE feature |
| Transluce-Explain [23] | Llama 3.1 8B Instruct | neuron |
| Llama Scope [27] | Llama 3.1 8B Base | SAE feature |
| Gemma Scope [45] | Gemma 2 | SAE feature |
| Goodfire SAE [29] | Llama 3.3 70B | SAE feature |
| Output-Centric neuron [25] | Llama 3.1 8B Instruct | neuron |
| Output-Centric SAE [25] | Gemma 2 2B, GPT-2 Small, Llama 3.1 8B | SAE feature |

## A.3 Description Scoring Details

Figure 7 illustrates the COSY evaluation procedure [39] as adapted for language models. As the control dataset $\mathbb{X}_0$, we use a subset of 1,000 randomly sampled entries from Cosmopedia [40]. For each candidate description of a target feature, we use Gemini 1.5 Pro [47] to generate 10 concept-specific text samples, each with a maximum length of 512 tokens. These samples form the concept dataset $\mathbb{X}_1$. The generation prompt is shown in Figure 8. We then pass both datasets through the model to extract activations corresponding to the target feature. We then use Average Pooling as aggregation function $\sigma : \mathbb{R}^d \to \mathbb{R}$ to each activation vector to obtain scalar representations:

$$
\begin{aligned}
\mathbb{A}_0 = \{\sigma(f_{\ell,i}(\boldsymbol{x}_1^0)), \ldots, \sigma(f_{\ell,i}(\boldsymbol{x}_n^0))\} \in \mathbb{R}^n, \\
\mathbb{A}_1 = \{\sigma(f_{\ell,i}(\boldsymbol{x}_1^1)), \ldots, \sigma(f_{\ell,i}(\boldsymbol{x}_m^1))\} \in \mathbb{R}^m.
\end{aligned}
\tag{8}
$$

The resulting activation distributions $\mathbb{A}_0$ and $\mathbb{A}_1$ are compared to compute the COSY Score (see Equations 6 and 7), which quantifies how accurately a given description captures the target feature.

Higher scores indicate clearer separation between concept and control samples, reflecting more precise and informative descriptions.

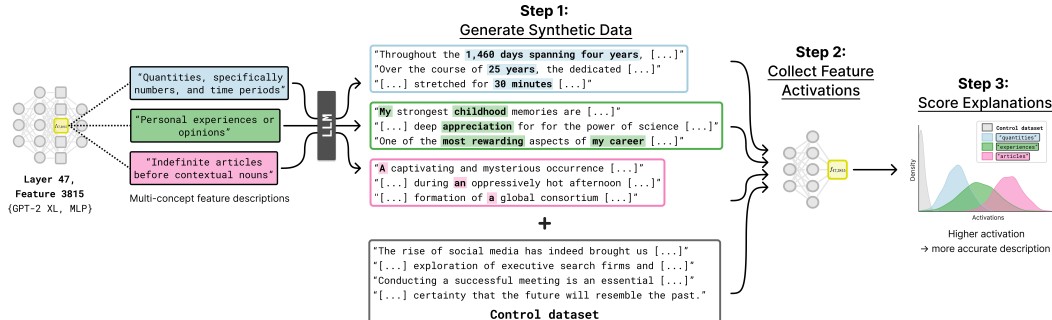

Figure 7: Score feature descriptions with CoSy. First, we compile multiple candidate descriptions for a target feature. For each description, we prompt an LLM to generate 10 text samples including the described concept. These concept-specific samples, along with a control set of random text samples, are processed through the model to extract activations for the target feature. The CoSy Score quantifies the separation between activation distributions of concept samples versus control samples, enabling objective comparison of different feature descriptions. Higher scores indicate descriptions that better capture the feature's underlying concept.

> Generate 10 sentences with a length of 512 words, one per line, with no additional formatting, introduction, or explanation. Each sentence should be a complete, standalone text sample that can be saved as an individual row in a text file. The sentences should include: `{feature_description}`

Figure 8: Prompt used to generate concept-specific text samples for evaluation. The placeholder `{feature_description}` is replaced with a candidate textual feature description before being passed to a large language model (Gemini 1.5 Pro). The model then generates 10 standalone text samples, which form the concept dataset $\mathbb{X}_1$. These samples are used to evaluate how well the description aligns with the target feature.

## A.4 Benchmark Experiment Details

**Reference Descriptions** We use the publicly available descriptions for GPT-Explain[9], Transluce-Explain[10], Neuronpedia[11], and Output-Centric[12] as comparison.

**Model Layers** For GPT-2 XL[13], we use layers 0, 20, and 40. For Llama 3.1 8B Instruct[14], we sample from layers 0, 20, and 30. For GPT-2 Small SAE, we use the original implementation[15], specifically version 5 with a width of 32k. We select features from layers 0, 5, and 10. For Gemma Scope[16], we use the residual stream SAE with width 16, selecting features from layers 0, 10, and 20.

**Feature Selection** For each model, we randomly choose 60 features, 20 from each of three layers, with available reference descriptions from prior work. The only exception is GPT-2 Small SAE, where only 59 features are annotated in the Output-Centric benchmark.

---

[9] `https://github.com/openai/automated-interpretability/tree/main?tab=readme-ov-file#public-datasets`

[10] `https://github.com/TransluceAI/observatory?tab=readme-ov-file#neuron-descriptions`

[11] `https://www.neuronpedia.org/`

[12] `https://github.com/yoavgur/Feature-Descriptions`

[13] `https://huggingface.co/openai-community/gpt2-xl`

[14] `https://huggingface.co/meta-llama/Llama-3.1-8B-Instruct`

[15] `https://github.com/openai/sparse_autoencoder?tab=readme-ov-file`

[16] `https://huggingface.co/google/gemma-scope`

**Prompt for generating Descriptions**  To produce textual feature descriptions, we use a prompt that instructs a large language model (Gemini 1.5 Pro) to identify shared concepts across high-activation text excerpts within a cluster. The model receives the top $N_s = 20$ excerpts per cluster, with high-activation token spans highlighted using square brackets. The full prompt is shown in Figure 9.

---

You are a meticulous AI researcher conducting an important investigation into a specific neuron inside a language model that activates in response to text excerpts. Each text starts with ">" and has a header indicated by === Text #1234 ===, where #1234 can be any number and is the identifier of the text.

Neurons activate on a word-by-word basis. Also, neuron activations can only depend on words before the word it activates on, so the description cannot depend on words that come after, and should only depend on words that come before the activation.

Your task is to describe what the common pattern is within the following texts. From the provided list of text excerpts, identify the concepts that trigger the activation of a particular feature. If a recurring pattern or theme emerges where these concepts appear consistently, describe this pattern. Focus especially on the spans and tokens in each example that are inside a set of [delimiters] and consider the contexts they are in. The highlighted spans correspond to very important patterns.

At the beginning, before the list of texts, there will be a list of the highlighted tokens with their activation values.

At the end, following 'Description:', your task is to write the description that fits the above criteria the best.

Do NOT just list the highlighted words!

Do NOT cite any words from the texts using quotation marks, but try to find overarching concepts instead!

Do NOT write an entire sentence!

Do NOT finish the description with a full stop!

Do NOT mention the [delimiters] in the description!

Do NOT include phrases like 'highlighted spans', 'Concepts of', or 'Concepts related to', and instead only state the actual semantics!

Do NOT start with 'Description:' and instead only state the description itself!

---

Figure 9: Prompt used to generate textual descriptions of a feature based on its activation patterns. The language model (Gemini 1.5 Pro) is instructed to analyze a set of text excerpts, focusing on highlighted spans corresponding to high activations of a specific feature. The model is guided to identify consistent patterns or concepts that trigger the feature. The resulting output is a concept-level description used as textual feature description.

**AUROC and MAD Distributions**  To better understand the standard deviation observed in our benchmark results, we provide distribution plots of the evaluation metrics. Figure 10 shows the distribution of AUROC scores across all evaluated model features, while Figure 11 presents the distribution of MAD scores.

**Output-Centric Evaluation**  In addition to our activation-based metrics, AUC and MAD, we evaluate an output-centric metric, Faithfulness, which quantifies the causal influence of a discovered concept on the model's output [81, 25]. Faithfulness measures the causal effect of a feature on model outputs, specifically testing whether directly manipulating a feature's activations can steer the model to generate content that more strongly reflects the corresponding concept. Following the FADE[17] implementation of [81], we compute Faithfulness scores of GPT-2 XL feature descriptions using GPT-4o mini [82] as the language model and a subset of 10,000 samples from the training split of the Cosmopedia dataset [40]. To obtain scores for all features, we remove the threshold used for selecting output-centric samples. Figure 12 presents the results of the output-centric evaluation. Across layer groups, PRISM (max) consistently outperforms GPT-Explain, exhibiting trends that closely mirror those observed in the activation-centric evaluation (see Figure 3(a)), further validating its effectiveness across multiple interpretability criteria.

---

[17]https://github.com/brunibrun/FADE

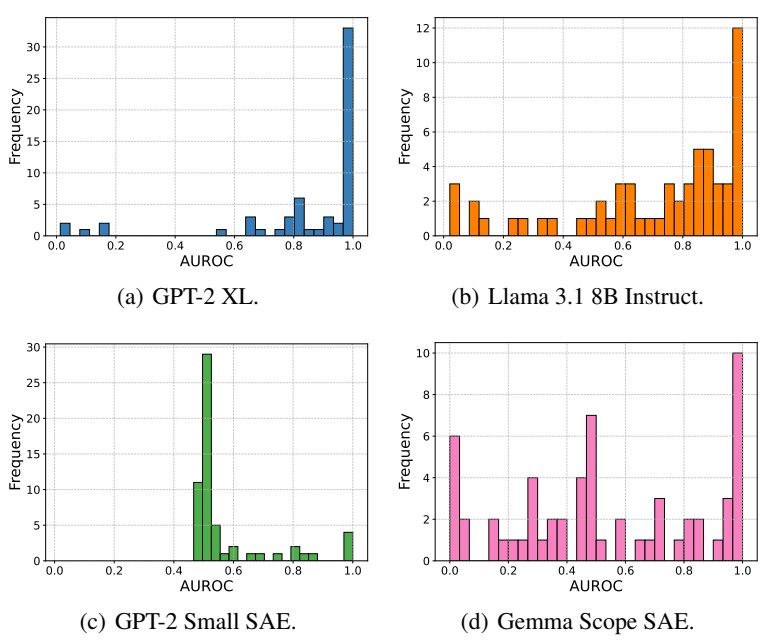

Figure 10: Distributions of PRISM (max) AUROC scores across different models and layers.

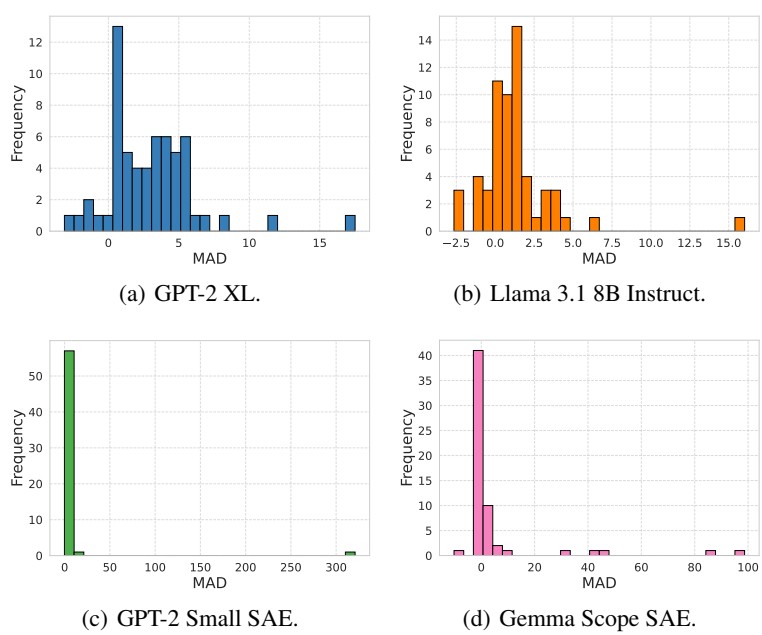

Figure 11: Distributions of PRISM (max) MAD scores across different models and layers.

**Compute Resources**    All experiments were conducted using a single NVIDIA A100 80GB GPU. The description procedure takes approximately 9 minutes per feature, including percentile sampling, clustering, and the generation of 5 descriptions. For evaluation, the generation of 10 sentences per feature requires roughly 3 minutes.

## A.5    Extended Ablation Analysis

**Impact of Varying Cluster Size**    We evaluate PRISM's performance across different numbers of clusters ($k$) for generating feature descriptions for GPT-2 XL. As shown in Table 3, increasing

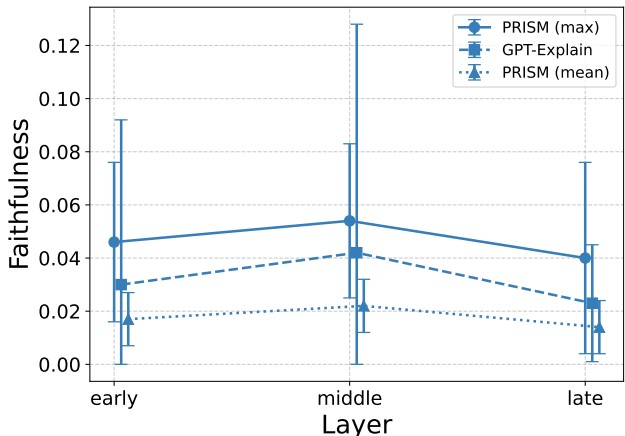

Figure 12: Output-Centric evaluation of GPT-2 XL across different feature description methods (GPT-Explain, PRISM (max), PRISM (mean)). Each method is evaluated using the output-centric Faithfulness metric, which quantifies the causal influence of a discovered concept on the model's output. PRISM (max) consistently achieves higher Faithfulness compared to GPT-Explain.

Table 3: Impact of the number of clusters ($k$) on feature description quality for GPT-2 XL. A cluster size of $k = 5$ corresponds to the original implementation used throughout the paper. Increasing $k$ improves best-case descriptions (higher PRISM max scores) by capturing more specific activation patterns, but lowers average interpretability (lower PRISM mean scores) as patterns fragment across clusters. This illustrates the tradeoff between precision and redundancy. AUROC values are reported with 95% confidence intervals; MAD values are reported as the percentage of positive MAD scores.

| $k$ Clusters | PRISM (max) | | PRISM (mean) | |
| --- | --- | --- | --- | --- |
| | AUROC ($\uparrow$) | MAD ($\uparrow$) | AUROC ($\uparrow$) | MAD ($\uparrow$) |
| 5 (original) | 0.85 (0.78-0.91) | 91.67% | 0.65 (0.61-0.69) | 66.33% |
| 1 | 0.75 (0.66-0.83) | 80.00% | 0.75 (0.66-0.83) | 80.00% |
| 3 | 0.82 (0.74-0.89) | 81.67% | 0.69 (0.64-0.74) | 68.33% |
| 10 | 0.88 (0.83-0.93) | 93.33% | 0.61 (0.58-0.64) | 59.67% |

$k$ improves best-case description quality (higher PRISM max scores), since clusters capture more specific activation patterns and yield sharper labels for the most coherent ones. At the same time, larger $k$ reduces average interpretability (lower PRISM mean scores), as coherent patterns become fragmented into statistically indistinguishable subclusters. These results illustrate the tradeoff: increasing the number of clusters enhances precision for the best descriptions but introduces redundancy and semantic overlap across clusters.

**Text Generators for Description Generation**   To test the robustness of our framework across different text generators, we extended the description generation experiments beyond Gemini 1.5 Pro to several open-source language models: Qwen3 32B [50][18], Phi-4 [51][19], DeepSeek R1 [52][20]. These models were used to generate feature descriptions for GPT-2 XL using the same procedure as in our original setup described in Section 4.1. As shown in Table 4, Qwen3 32B achieves performance comparable to Gemini 1.5 Pro, demonstrating that our framework does not depend solely on a single model. While Phi-4 and DeepSeek R1 show slightly lower scores, they still follow the same qualitative trends, demonstrating that our method is robust and generalizes effectively across a range of language models, including accessible open-source options.

**Text Generators for Evaluation**   We evaluated the robustness of our feature descriptions by using Qwen3 32B [50], Phi-4 [51], DeepSeek R1 [52] as alternative text generators in the evaluation

---

[18]https://huggingface.co/Qwen/Qwen3-32B

[19]https://huggingface.co/microsoft/phi-4

[20]https://huggingface.co/deepseek-ai/DeepSeek-R1

Table 4: Ablation study on text generators used for description generation. We report PRISM scores (max and mean) for different LLMs using the original experimental setup. Qwen3 32B achieves performance comparable to Gemini 1.5 Pro, while Phi-4 and DeepSeek R1 exhibit slightly lower scores but maintain the same qualitative trends, demonstrating the framework's effectiveness across multiple language models. Reported AUROC includes 95% confidence intervals; reported MAD represents the percentage of positive MAD scores.

| Text Generator (description) | PRISM (max) | | PRISM (mean) | |
|---|---|---|---|---|
| | AUROC (↑) | MAD (↑) | AUROC (↑) | MAD (↑) |
| Gemini 1.5 Pro [47] (original) | 0.85 (0.78-0.91) | 91.67% | 0.65 (0.61-0.69) | 66.33% |
| Qwen3 32B [50] | 0.85 (0.78-0.91) | 90.00% | 0.65 (0.61-0.69) | 64.33% |
| Phi-4 [51] | 0.82 (0.75-0.89) | 85.00% | 0.61 (0.57-0.65) | 61.33% |
| DeepSeek R1 [52] | 0.79 (0.71-0.87) | 78.33% | 0.61 (0.56-0.65) | 60.33% |

Table 5: Ablation study on text generators used in the evaluation step. We report AUROC and MAD scores for PRISM and GPT-Explain when using Gemini 1.5 Pro (original setup) or alternative LLMs (Qwen3 32B, Phi-4, DeepSeek R1) for description scoring. While absolute scores decrease with alternative models, relative rankings remain stable, with PRISM consistently outperforming GPT-Explain. AUROC is reported with 95% confidence intervals, and MAD is reported as the percentage of positive MAD scores.

| Text Generator (evaluation) | PRISM (max) | | PRISM (mean) | | GPT-Explain | |
|---|---|---|---|---|---|---|
| | AUROC (↑) | MAD (↑) | AUROC (↑) | MAD (↑) | AUROC (↑) | MAD (↑) |
| Gemini 1.5 Pro [47] (original) | 0.85 (0.78-0.91) | 91.67% | 0.65 (0.61-0.69) | 66.33% | 0.64 (0.56-0.73) | 65.00% |
| Qwen3 32B [50] | 0.58 (0.46-0.69) | 58.33% | 0.53 (0.48-0.58) | 54.00% | 0.54 (0.42-0.65) | 53.33% |
| Phi-4 [51] | 0.61 (0.50-0.72) | 58.33% | 0.54 (0.49-0.59) | 53.67% | 0.56 (0.44-0.67) | 58.33% |
| DeepSeek R1 [52] | 0.71 (0.61-0.80) | 73.33% | 0.57 (0.52-0.62) | 57.67% | 0.60 (0.50-0.70) | 63.33% |

step (Appendix A.3 provides details on description scoring). In this process, a set of 10 concept-specific text samples for each feature is generated with the LLM, and AUROC and MAD scores are computed to assess description quality for both PRISM and GPT-Explain (Table 5). Although absolute scores are slightly lower when Gemini 1.5 Pro is not used for evaluation, the relative ranking of methods remains consistent. PRISM consistently outperforms GPT-Explain, demonstrating the generalizability of our framework and the robustness of our evaluation setup.

## A.6 Extended Sanity Check Analysis

**Random Sentences in Clusters** To ensure that the generated feature descriptions are faithful to the tokens and corresponding text samples in the same cluster, we perform a fully randomized counter probe. For this sanity check, we replace the percentile sampling in Step 1 of Figure 2 with random sampling, drawing random sentences and their corresponding activations from the validation set of the C4 CORPUS [41]. Since choosing highly activating samples across our random set would bias the random uniform sample distribution, we apply the clustering procedure across all random samples. While we maintain the cluster size, all text samples and highlights are shuffled and assigned to random clusters instead. Both the assignment of random clusters and highlights limits sample similarity. Lastly, we proceed with labeling the randomized clusters. Each cluster consists of unrelated texts and highlights, thus, the LLM should be unable to generate a coherent, shared feature description. As shown in Table6, AUROC and MAD scores decrease, confirming that meaningful descriptions depend on the proper grouping of highly activating samples.

**Random Descriptions Experiment** We probe whether our feature descriptions truly capture the concepts of their assigned clusters by performing an additional randomization of descriptions. We reassign feature descriptions by randomly sampling from existing descriptions generated for other features within the same model and collect the associated activations. This tests whether explanation quality decreases when descriptions no longer correspond to their clusters, without generating new descriptions. To this end, we use the same feature and layer selection as described in Section 4.1 for GPT-2 XL, which serves as the baseline for non-randomized PRISM performance (see Table 1 GPT-2 XL, MLP neuron). We expect AUROC and MAD scores to drop significantly, since the descriptions do not align with the cluster. Table 6 shows that AUROC and MAD scores drop significantly

Table 6: Sanity check results for GPT-2 XL. We compare AUROC and MAD scores under two randomized settings: (1) Random Sentences, where clusters are formed from unrelated text samples, and (2) Random Descriptions, where existing descriptions are randomly reassigned across features. Both conditions show a notable drop in AUROC and MAD scores compared to the Baseline, supporting the faithfulness of our descriptions. AUROC reports $95\%$ confidence intervals; MAD shows the percentage of positive MAD scores.

| | PRISM (max) | | PRISM (mean) | |
| --- | --- | --- | --- | --- |
| | AUROC ($\uparrow$) | MAD ($\uparrow$) | AUROC ($\uparrow$) | MAD ($\uparrow$) |
| Baseline | 0.85 (0.78-0.91) | 91.67% | 0.65 (0.61-0.69) | 66.33% |
| (1) Random Sentences | 0.68 (0.59-0.76) | 65.00% | 0.54 (0.50-0.58) | 49.67% |
| (2) Random Descriptions | 0.65 (0.56-0.74) | 66.67% | 0.52 (0.48-0.57) | 49.33% |

compared to the baseline, demonstrating that aligned descriptions are essential for high-quality feature description.

**Random Descriptions Polysemanticity Scores Comparison**  To verify that polysemanticity scores reflect meaningful semantic relationships rather than artifacts of the embedding process, we perform a sanity check using randomly assigned descriptions. For each feature, five random descriptions (distinct from the true set) are embedded using the same embedding model (gte-Qwen2-1.5B-instruct sentence transformer [46]), and pairwise cosine similarities are computed between them. This is repeated across all features within each layer, and the resulting scores are visualized as box plots grouped by layer (see Figure 13). The difference between true and random scores is generally quite distinct, with random descriptions yielding notably lower similarity values, indicating less semantic coherence. An exception is Llama 3.1 8B Instruct, where cosine similarity scores remain consistently low across layers and frequently approach the random baseline. This indicates that the descriptions associated with Llama's features are likely more diverse and less semantically aligned compared to other models.

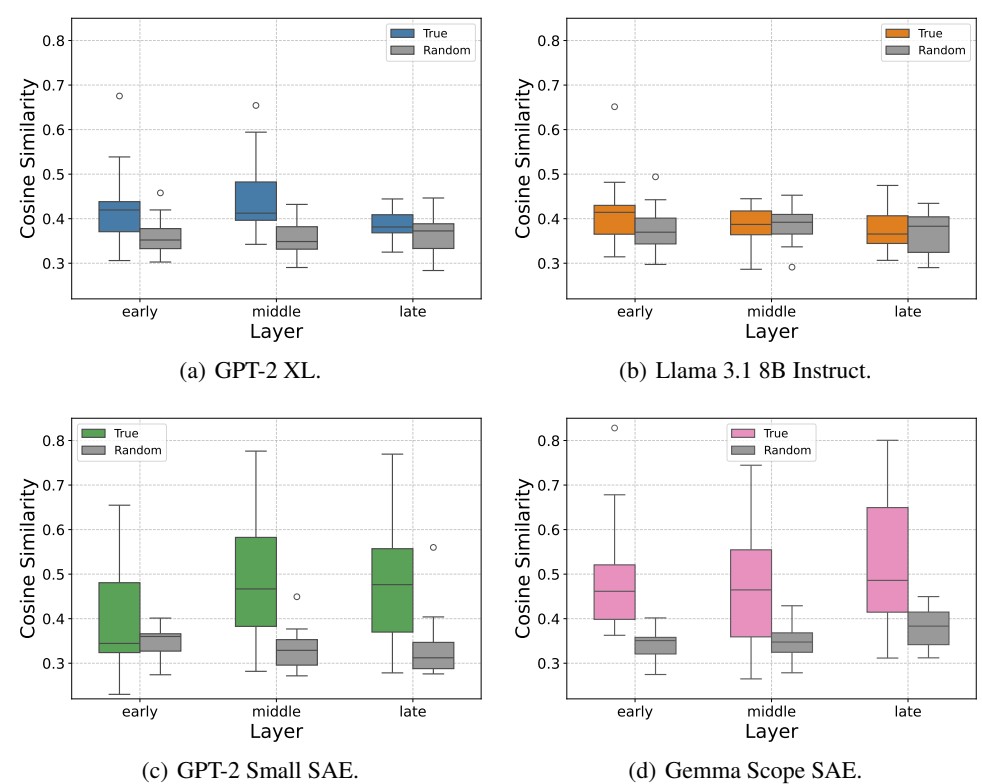

(a) GPT-2 XL.

(b) Llama 3.1 8B Instruct.

(c) GPT-2 Small SAE.

(d) Gemma Scope SAE.

Figure 13: Comparison of true and random polysemanticity scores across models and layer groups.

Table 7: Performance across percentile intervals of feature scores on GPT-2 XL. AUROC measures classification performance (95% CI in parentheses), and MAD quantifies activation differences (percentage of positive MAD scores). Results are shown for both PRISM (max) and PRISM (mean).

| Intervals | PRISM (max) | | PRISM (mean) | |
|---|---|---|---|---|
| | AUROC ($\uparrow$) | MAD ($\uparrow$) | AUROC ($\uparrow$) | MAD ($\uparrow$) |
| Baseline | 0.85 (0.78-0.91) | 91.67% | 0.65 (0.61-0.69) | 66.33% |
| 0.0 to 0.25 | 0.69 (0.61-0.77) | 66.67% | 0.53 (0.49-0.57) | 49.33% |
| 0.25 to 0.5 | 0.74 (0.66-0.82) | 70.00% | 0.56 (0.52-0.61) | 52.67% |
| 0.5 to 0.75 | 0.71 (0.63-0.79) | 73.33% | 0.55 (0.51-0.60) | 54.33% |
| 0.75 to 1.0 | 0.85 (0.79-0.91) | 90.00% | 0.65 (0.61-0.69) | 66.33% |

**Percentile Interval Analysis**  We examine whether feature description quality varies across the percentile activation distribution by segmenting the distribution into quartile-based intervals. For consistency, we apply the same PRISM parameter settings as in the main benchmarking experiments, varying only the start and end points of the percentile range. The baseline setting uses the top 1% of activations (0.99-1.0). Table 7 reports AUROC and MAD scores across intervals. As expected, features in the top 25% (0.75-1.0) closely match baseline performance, while lower intervals exhibit reduced scores, indicating a positive correlation between activation strength and description quality.

**Relative Activation Analysis**  To ensure that feature descriptions are based on relevant and representative samples, we analyze the relative activations between the 99th and 100th percentile samples for each neuron. For a neuron $j$ in a given layer and dataset, let $a_{\max}^j$ denote the maximum (100th percentile) activation and $a_{99}^j$ the 99th percentile activation. We define the relative ratio

$$r_j = \left| \frac{a_{99}^j}{a_{max}^j} \right| \in [0, 1]. \tag{9}$$

We evaluate $r_j$ across three layers (0, 20, and 40) of GPT-2 XL (see Figure 14), using the same settings as in our main experiments. In Layers 0 and 20, no neurons exhibit $r_j < 0.15$. In Layer 40, the ratios are generally lower, but none fall below $0.05$. This indicates that extremely small relative activations are rare, particularly in early layers. Furthermore, across layers, our method is able to robustly extract feature descriptions, even for neurons with more diverse $r_j$ values (see also Figure 3(a)).

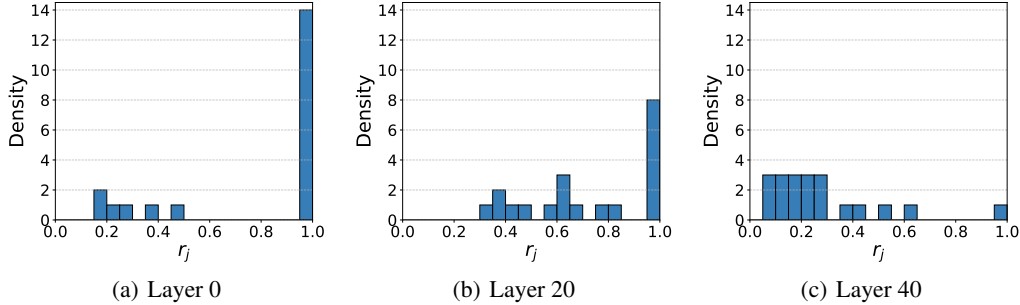

(a) Layer 0        (b) Layer 20        (c) Layer 40

Figure 14: Distribution of the relative activation ratio $r_j$ between the 99th and 100th percentile samples across three layers (0, 20, 40) of GPT-2 XL. This ratio measures how representative the top neuron activations are compared to near-peak activations. Ratios in Layers 0 and 20 are consistently higher, while Layer 40 shows lower values but none below 0.05, indicating that extreme outlier activations are rare.

To further illustrate that sampling from diverse activation ranges is meaningful in practice, we present a selection of feature descriptions. Table 8 shows cases where clusters with diverse mean activation ranges (MeanAct) still achieve high description scores (AUROC) (also see Figure 1).

Table 8: A selection of feature descriptions obtained from clusters with diverse mean activation ranges (MeanAct). Despite variation in activation ranges, the descriptions achieve consistently high scores (AUROC), illustrating that percentile sampling captures multiple meaningful patterns per feature.

| Model, layer, feature | Description | MeanAct | AUROC |
|---|---|---|---|
| GPT-2 XL, 47, 3815 | Quantities, specifically numbers, and time periods | 0.46 | 0.99 |
| | Personal experiences or opinions | 0.42 | 0.98 |
| | Indefinite articles preceding nouns related to events, times, groups, places, and objects | 0.75 | 0.99 |
| Llama 3.1 8B Instruct, 30, 6472 | End-of-sequence tokens following descriptions of services, products, or academic programs | 0.35 | 0.99 |
| | Locations, events, or entities and their associated details | 0.22 | 0.90 |
| | Possessives or the beginning of numbered lists | 0.22 | 0.85 |
| GPT-2 Small, 10, 4369 | Financial institutions, people's names, and technical terminology related to medical scans | 3.08 | 0.58 |
| | A person's name containing "ib" | 4.55 | 0.89 |
| | International Business Machines (a technology company) and cystic fibrosis | 2.95 | 0.99 |
| Gemma Scope, 0, 12182 | Products or items and their specifications or descriptions | 1.33 | 0.84 |
| | Home improvements, dental procedures, taxes, and savings accounts | 1.56 | 1.0 |
| | Brexit negotiations/deals | 2.00 | 0.69 |

The analysis of relative activations in both percentile sampling and clustering suggest that percentile sampling reflects a focus on diverse functional and meaningful patterns when present. This aligns with our goal of retrieving multiple distinct patterns per feature, resulting in multiple descriptions per feature.

### A.7 Metalabels

In Figure 15 and Figure 16, we provide additional examples of metalabels for GPT-2 XL and GPT-2 Small SAE. These resulted from clustering 300 sentence representations (embedder: GPT-2 XL, last-token pooling) of identified feature descriptions for a given model and neurons. We show 20 randomly selected samples from a total of $k_m = 50$ meta-clusters that were computed using k-means, along with up to three feature descriptions selected at random. Metalabel descriptions were generated via Gemini 1.5 Pro. Clusters for which no concise label was generated, are labeled with 'N/A'.

As discussed in Section 5.1 for GPT-2 XL, similar patterns are observed for GPT-2 Small SAE (see Figure 16), including semantic categories like "Spatiotemporal Descriptions and Personal Anecdotes" (id 20), syntactic concepts like "Pattern Matching" (id 22), and task-specific representations such as "Instructional/Explanatory" (id 49) and domain-focused clusters like "Publication and Distribution" (id 35). Of particular interest are task- and domain-specific examples, such as "Listeria Contamination" (id 46), which suggest concepts linking semantics with pragmatics. These concepts convey context-specific intents, such as warning or ensuring public safety, highlighting how pragmatic factors influence model interpretation.

### A.8 Human Study Details

**Participants** Seven participants from two different academic institutions took part in the study. All were either PhD students or working students and were compensated for their participation. None were co-authors or otherwise involved in the project. Completing the survey required an average of 140 minutes.

**Study Setup** Each participant was presented with 8 groups of sentence clusters, where each group corresponded to one feature and consisted of 5 clusters of highly activating sentences (20 sentences per cluster), along with highlighted tokens. For each cluster, participants were tasked to write a short textual description, resulting in 5 human-generated descriptions per feature. They then rated the pairwise similarity between these descriptions on an 11-point scale (0.0–1.0), where 1.0 indicates very high similarity. The instructions provided to participants are shown in Figure 17.

**Results** Figure 6 presents the results, sorted by PRISM score (lower values indicate higher polysemanticity). We report the average of these ratings as the Human polysemanticity score. In addition, we compute Human Cosine Similarity using the same embedding model (gte-Qwen2-1.5B-instruct sentence transformer [46]) and method used for computing the PRISM polysemanticity score, with the only change being the use of human-generated descriptions instead of model-generated ones. As expected, features with low PRISM scores receive lower human similarity ratings and lower embedding-based similarity. For example, the feature from GPT-2 Small (layer 5, feature 30319) has a low PRISM score (0.30), and is judged by participants to be semantically diverse (Human

polysemanticity score: 0.40). In contrast, features with high PRISM scores, such as those from Gemma Scope (features 10531 and 603), show strong human agreement (Human polysemanticity score: 0.90 and 0.80, respectively) and high Human Cosine Similarity (0.79 and 0.75). These findings provide both qualitative and quantitative support for the PRISM metric, showing that it aligns well with the human interpretation of polysemanticity.

| id | Metalabel | Feature Descriptions |
|---|---|---|
| 3 | Technology and Specifications | - Settings, assignments, or actions related to software or applications
- Textile material, food and beverage, medicinal substances
- Qualities, characteristics, or specifications of animals or products |
| 4 | Online Discourse and New Experiences | - A first time experience, often with an element of surprise or anticipation
- Social media, Donald Trump, Twitter, counsel, upbeat tweets
- Apologies, add-ins, testaments, descendants, grades, versions, dialogue, honesty, downgrades, payments, relevance, sensibility, timelessness, credit, superiors, decency, hardened hearts, ageing, genuinely frightened by reality, reliability, shrouded, credited, ironically permitted, gigs, obsession, ... |
| 6 | Positive Experiences | - Expressions of excitement, sharing, or positive feedback
- Expressions of gratitude, current time references, or positive descriptions
- Experiences related to travel, leisure activities, meals, and events, particularly those with a temporal element (time, dates, or duration) |
| 7 | Commerce/ Finance | - Months, numbers, and second-hand collectibles, rentals, applications, retail spaces, or holiday gifts
- Holiday or special occasion accessories/decorations
- Food, tools/devices, and cosmetics/accessories |
| 9 | Access/ Acquisition | - Transfer, storage, or placement of objects or people
- Consumption of food, beverages, or medications, sometimes for free or at a reduced price
- Winning a prize or participating in a competition |
| 13 | Personal and Professional Experiences and Standards | - A discussion of customer service experiences, religious figures and texts, TV series, sporting events, community events, restaurants, international summits, golf rules, summer camps, company performance, and international relations
- Proper nouns, often people's names, in contexts of competitions, scandals, or events, especially when related to games, sports, politics, or entertainment
- Professional standards, requirements, and practices related to a variety of fields, including surveying, reviewing products/services, nutrition, data analysis, education, healthcare, music, and spirituality |
| 14 | Achievements, Solutions, Lifestyle, Risks, News and Controversy, Conflicts, Corruption | - Events, particularly those related to conflict, competition, or problematic situations
- Topics related to politics, sports, and current events, specifically focusing on major decisions, outcomes, and controversies
- Government-related scandals and investigations, particularly those involving leaks, cover-ups, or accusations of wrongdoing |
| 18 | Structured Data | - Numerical or quantitative values, including years, measurements, and counts, in advert-like text excerpts
- Titles, names, locations, and dates in bibliographic entries or citations
- Academic degrees, professional roles, locations, and years related to education or employment history |
| 19 | Development and Improvement | - Mental and/or physical manipulation or transformation
- Funding of projects and initiatives related to healthcare, social issues, and education
- Methods, procedures, or sequences related to improvement, change, or progression |
| 21 | Personal Reflections | - First-person accounts, often expressing personal opinions, beliefs, or experiences
- Personal updates
- Experiences, actions, and feelings related to entertainment, media, and technology, along with personal anecdotes or opinions |
| 26 | Products/ Services and Medical Information | - Products or services with descriptions and/or characteristics
- Products or services with details or instructions
- Medical conditions, types of medical treatment, or medical professionals, sometimes involving a duration or repeating pattern |
| 27 | N/A | - Fitness, essays, digital skills training, product design and marketing, audits, internships, coaching, graphic design, academic assistance, data analytics, predictive modeling, computational chemistry, handwriting development, software documentation, development tools, intellectual property, ...
- Business services, including career services, company recruitment, tours, and consulting, offered by organizations, for students and professionals, in various fields, such as marketing, technology, and healthcare
- Rope access, locations in Arkansas, educational courses, probation terms, Nigerian aid, hospital communication improvement, South American airline travel, Nigerian profession improvement, admission to a school nursery, Florida ... |
| 28 | Business and Organization Information | - Locations of businesses or organizations
- Geopolitical events and entities involved, particularly government actions and agencies
- Events, services, or products offered by a business or organization |
| 29 | Events and Activities | - Achievements, awards, or special events, often including a specific person or group
- Food, specific locations, and activities/routines, often involving a change in direction or state
- Events, shows, or locations, often with a time or date, and sometimes including named people |
| 35 | Personal Development and Management | - Self-awareness, identity, and reality, often related to technology and its impact on the user
- Discussions of financial, life, or career planning, resource allocation, and caregiving, often in the context of family or children
- Actions or states of being, often ongoing or recently completed |
| 38 | Diverse Inquiries and Services" | - Promotional products or gifts relating to cuteness and popularity with customers
- Locations, often neighborhoods or districts, and named entities associated with those locations
- French language learning, professional certifications and qualifications, and company services related to specific industries |
| 42 | N/A | - First-person introspection, often related to mental and emotional states, self-awareness, and personal beliefs
- First-person perspective related to identification, personal information, and objects
- Conditional actions or situations and their potential outcomes, especially relating to rules, regulations, or personal choices |
| 44 | N/A | - Legal cases, particularly theft and court proceedings, and discussions of sports teams and players
- International trade, diplomacy, and agreements between countries
- Past events, especially from a year ago or more |
| 45 | Legal and Administrative Affairs | - Division of assets/property, medical procedures/treatments, legal disputes/court proceedings, and organizational activities/events
- Ownership of property or membership status
- Commercial transactions, legal proceedings, and financial obligations |
| 48 | N/A | - Health benefits of avocado, color testing tools, gene normalization for hPDL fibroblasts, teriyaki chicken wings, reptile hemoparasite identification, hemorrhoid treatments, omega-3 fatty acid supplements for fertility, baking trout, ...
- Activities related to hobbies including airplane livery design, scrapbooking, SMART team coordination, journaling, early childhood sensory play and development, painting vegetables, taking consumer surveys, decorating with lampshades, ...
- Infants, food, and crafting |

Figure 15: Clustering of identified PRISM feature descriptions in **GPT-2 XL**. Shown are the $k = 50$ meta-clusters of feature descriptions, each labeled with a corresponding metalabel generated by Gemini 1.5 Pro, along with up to 3 randomly selected sample descriptions per cluster.

| id | Metalabel | Feature Descriptions |
|---|---|---|
| 3 | Miscellaneous | - Japanese teriyaki chicken, a type of sedan, art pieces featuring wood, sugary food/drinks, gameplay mechanics, leggings, plumbing services, file names of furniture images
- Hib vaccine, investor-owned utilities, couples counselling, weak economy/immune system, HARPO fellowship, outage
- Energy-related industry or resource |
| 4 | Seeking and Managing Resources | - Data storage, transfer, or management, often in relation to websites, software, or online platforms
- Ordering, requesting, or discussing types of services, accounts, or information, often related to online platforms, finances, or businesses
- Requesting, searching for, or looking for something, especially services like legal or insurance, or items like quotes or properties |
| 6 | Linguistic Elements and Structures | - The conjunction "So" starting a sentence, often introducing a conclusion or consequence based on the preceding context
- Conjunctions, prepositions, and occasionally other function words, appearing in descriptions of food and drink preparation, achievement announcements, or product descriptions
- Commercial enterprises, locations of residence, textual works, family members, sports, and proper nouns |
| 8 | N/A | - Occupations or roles related to water
- Locations (cities, states, or neighborhoods) and things found in homes or related to home maintenance/improvement
- Products or services related to attire, beauty, or personal care |
| 12 | Products and Locations | - Clothing, accessories, or cosmetic products with descriptions of their materials, features, or benefits
- Items or products and their descriptions including specifications, materials, and uses
- Medical and/or chemical terms in the context of product descriptions or technical documentation |
| 17 | Competitive Analysis | - Business competitors/competition
- Evaluation, academic/educational institutions, and certification/qualification
- Publication details, often including author, title, date, and publisher/journal |
| 19 | N/A | - IndyCar series or races, often with the word "Indy" highlighted
- Questions about processes, mostly using the auxiliary "does"
- A person named Fei appearing in a conversational context |
| 20 | Spatiotemporal Descriptions and Personal Anecdotes | - Locations, proper nouns, and numbers related to places, events, or entities
- Timestamps, specifically times of day
- A short personal story often including a mention of a family member, sometimes in relation to a specific past time or recent event |
| 22 | Pattern Matching | - The letter G, capitalized or not, related to proper nouns in a list-like structure
- A substring "ib" or "ibr", often within a proper noun, especially a person's name
- Names, punctuation marks, and specifically the tokens "sh", "!", "'", ")", and a newline character |
| 26 | Product/ Service Descriptions | - New service/product offerings or marketing/promotion of existing services/products
- Belonging, origins, sources, or components
- Product features or qualities |
| 27 | Ordinal Numbers | - Ordinal numbers in contextual information describing locations, groups, or ordered lists
- Ordinal numbers, often within the context of lists or ordered items
- Ordinal numbers in a numbered list |
| 28 | WordPress and Digital Business Skills | - Content related to the Wordpress platform, possibly focusing on its usage, features, and user groups
- Financial/business topics, digital skills training, or software platforms and their features/benefits
- Computer/IT skills, software, or computer programs, often in a business/professional/marketing/sales context |
| 34 | Digital Business and Technology | - Products and services related to pet care, home improvement, electronics, and computing
- Software, tools, and resources for creating and managing websites and other digital content
- Website/software development, marketing, and financial services/products |
| 35 | Publication and Distribution | - Relating to an edition or version of a book or relating to a card game
- Giveaway, donation request, advertisement, or sharing information, related to a link or media
- Relating to the beginning section of a piece of writing |
| 38 | N/A | - Medical studies of the effects of various factors or substances on different types of cancer
- Energy sources, including renewable energy like tidal power as an alternative to fossil fuels, geological formations and processes like sediments, and motor neuron degeneration
- International Business Machines (a technology company) and cystic fibrosis |
| 42 | N/A | - Locations offering services or events
- Competition, playoff, or tournament sporting events
- Discussions of computer hardware and software, website creation and management, recipes, personal anecdotes and hobbies, product reviews, and summaries of events |
| 44 | Product and Service Specifications | - Video file sharing, software, or online services related to media, including video resolution, file formats, platforms, and user experience
- Attributes of products related to materials, sizes/dimensions, and/or color
- Screen resolution or magnification |
| 46 | Listeria Contamination | - Food recalls due to bacterial contamination, specifically Listeria
- Food contamination with Listeria |
| 47 | Taxes and Legal Obligations | - Ownership of creative digital content, specifically relating to Italian architecture or fashion accessories and their online availability
- Tax obligations for non-resident sellers of real estate, especially focusing on the buyer's responsibility to withhold a percentage of the sale price for tax purposes
- Instructions related to food preparation, especially baking or chilling in a refrigerator, sometimes followed by serving instructions |
| 49 | Instructional/ Explanatory | - Demonstrative pronouns (this, that) at the beginning of sentences, especially related to new information or summaries
- Competitive situations, often sports or games, with emphasis on positions and actions taken by a team or individual player
- Second-person pronouns in instructional or user-manual style texts, frequently appearing in contexts involving explanations of processes, tools, or options available to the reader |

Figure 16: Clustering of identified PRISM feature descriptions in **GPT-2 Small SAE**. Shown are the $k = 50$ meta-clusters of feature descriptions, each labeled with a corresponding metalabel generated by Gemini 1.5 Pro, along with up to 3 randomly selected sample descriptions per cluster.

Large Language Models (LLMs) have rapidly become integral to a range of real-world applications, from software development to medical diagnostics. Despite their growing influence, the internal decision-making processes of these models remain largely opaque. We aim to understand these black-box systems by analyzing their internal structure.

In this survey, we're looking at **activation scores**, continuous values that can be both negative or positive, indicating how much a neuron (single unit of a neural network) in a LLM "reacts" to some input text at inference time. This has been calculated for each token (single unit of text). If an activation is positive (above zero), we consider that a **highlight**.
Neuron activations can only depend on words before the word it activates on, so the description cannot depend on words that come after.

For each of eight groups, you will see five clusters containing some number of **text samples**. The text samples all start with ">" and have a header indicated by "=== Text #1234 ===", where "#1234" represents the ID of the text sample.
Each **cluster** starts with a summary of the highlights (tokens with positive activations) alongside their activation scores. This summary is followed by all highlights in the context of the samples.

Your tasks:

(1) Describe what the **common pattern** is within the following texts. From the provided list of text excerpts, identify the **concepts** that trigger the activation of a particular feature. If a recurring pattern or theme emerges where these concepts appear consistently, describe this pattern. Focus especially on the spans and tokens in each example that are inside a set of [delimiters] and consider the contexts they are in.
(2) At the end of a group, **rate** the **perceived similarity** on an 11-point scale (0 to 10).
(3) At the end of the entire questionnaire, mention the **time** you took in total.

Remember these caveats for assigning descriptions:

\* Do not just list the highlighted words!
\* Do not write an entire sentence!
\* Do not finish the description with a full stop (".")!
\* Do not include phrases like 'highlighted spans', 'Concepts of', or 'Concepts related to', and instead only state the actual semantics!

Example:

CLUSTER #0: *Duration of service/contract/agreement, or time period of a ban/study/investment, especially in relation to business, finance, legal or technical contexts*
CLUSTER #1: *Exclusivity, short durations, or small quantities, sometimes referring to locations with "Spa" in their name*
CLUSTER #2: *Exclusivity, limitations, or allowances*
CLUSTER #3: *Intelligence gathering, growing plants, placing objects, time periods, marijuana use, higher goals, something only happening once, running and the greeting "hi", being originally from somewhere*
CLUSTER #4: *Exclusivity, allowance, placement, increased quantity/quality, and artistic activities*

Semantic similarity score: 6

Figure 17: Human user study instructions for annotating text clusters and rating the polysemanticity of their annotations.

