# OpenReview forum: "Capturing Polysemanticity with PRISM: A Multi-Concept Feature Description Framework"
_NeurIPS.cc/2025/Conference — NeurIPS 2025 poster_

### Official Review · Reviewer_9BmE · 2025-06-30

**Clarity:** 3
**Significance:** 2
**Originality:** 3
**Rating:** 4
**Confidence:** 4

**Summary:**

The paper proposes PRISM, an automatic framework to explain the features recognized by neurons in LLMs. Specifically, PRISM focuses on highlighting polisemanticity of neurons (the extent to which a neuron encodes multiple distinct concepts).

The framework works by first extracting the top percentile of activations for a given neuron and the samples (sentences) associated with that percentile. Then, the framework encodes these samples using a sentence encoder and clusters the generated embeddings via K-Means. Finally, the identified clusters are fed to an LLM, whose task is to assign a description to each cluster. The set of descriptions represents the "features" captured by the given neuron. To measure the degree of polysemanticity, the authors propose a new score, named Polysemanticity Score, and use the Mean Action Difference and AUC scores, which were originally proposed by the CoSy evaluation method for the vision domain.

PRISM is compared against GPT-Explain for explaining GPT-2 X neurons and against the Output-Centric method on three other LLMs using these metrics, achieving, most of the time, higher scores. Using the proposed framework, the paper also analyzes the concept space learned by a couple of LLMs, showing how the proposed framework can support different explanation granularities.

**Questions:**

See the section Weaknesses and Strengths.

Additionally, below you can find a question that didn’t have any impact on my evaluation, but it is related to the concern about the wording and I feel that including some additional information related to it into the main text could improve the informativeness of the paper:

- Could authors elaborate more on the sentence in line 28-30 *"While several concept extraction techniques like sparse autoencoders (SAEs) [17 , 18 ] aim to disentangle polysematic features, many resulting descriptions still reflect multiple concepts [19], limiting interpretability.)?"* What is the limitation of these techniques?

**Ethical Concerns:**

["NO or VERY MINOR ethics concerns only"]

**Final Justification:**

In my initial review, my concerns were related to the following points:
- A flaw in the evaluation setup (using the same model for both evaluation and optimization)
- Lack of ablation studies on some design choices
- Concerns regarding the metrics used to measure explanation quality
- Limited performance comparison (only one competitor per setup)
- Overstatements about the research niche addressed and the contribution
- Overstatement about the user study, which was initially performed with only one participant.

During the rebuttal phase, the authors:
- Fully addressed the concern regarding the ablation study
- Fully addressed the flaw in the evaluation setup (by providing results using different combinations)
- Addressed the concerns about the evaluation metrics
- Proposed changes to mitigate the overstatements about the niche and contribution
- (Partial) Expanded the user study to include 7 participants
- (Partial) Added one more competitor in the performance evaluation.


Overall, these improvements justify an increase in my score. I now consider the paper borderline but leaning towards acceptance. The main remaining weaknesses are the limited user study (only 7 participants) and the restricted performance evaluation (only 2 competitors per setup), despite the availability of many relevant methods in the literature. However, given the relevance of the topic, the flexibility and potential of the proposed method as a baseline in the field, and the significant improvements made in response to the review, I lean toward acceptance.

**Limitations:**

yes

**Paper Formatting Concerns:**

No paper formatting concerns

**Quality:**

1

**Strengths And Weaknesses:**

**Strengths**:
- **Significance and Originality**:

   - The *topic* of automated interpretability for LLMs is gaining interest lately. A contribution in the area could be useful for other researchers.
    - *Scientific merit* / method: I appreciate the general simplicity of the framework. Given the small time to compute the explanations and the relative easiness of implementation, I believe this framework could easily become one popular baseline for this research niche if the framework is associated with the right setup (e.g., open models, see "weaknesses, experimental setup" below).
   - While the single elements of the framework have been used in the literature (e.g., the combination of clusters and percentile or clusters and LLMs have been used in Vision to capture polysemanticity) their combination for this specific niche is *novel* to the best of my knowledge.
   - *Flexibility*: The framework supports explanations at different levels of granularity. The authors show that the framework is reliable in producing both high level explanations (meta-level/topics) and more fine-grained ones (feature descriptions). This is an important feature of the framework and one of its strongest points, in my opinion.

- **Clarity**: The paper is overall well structured and easy to read. There are no major concerns here.


**Weaknesses**:

All the below weaknesses are related and impact the *quality* score.

- **Experimental setup**:
   - Method: all the experiments are performed using Gemini Pro 1.5 to generate feature descriptions and using a specific prompt and a specific number of samples (20) to generate the feature descriptions. There is no ablation study on the configuration. Within this setup, it becomes difficult to understand whether the results depend on the specific configuration and LLM used or reflect some general capabilities, as a framework would require.  Similar to other points of this review, I would suggest either changing the wording and the narrative of the text to be more aligned with the experimental setup (e.g., positioning the paper as a framework that specifically exploits Gemini Pro) or (much better) provide evidence that the results are robust across different LLMs. In my view, the difference is substantial: it distinguishes between a paper proposing a method/framework and one that exploits a specific LLM to achieve a given goal. In this context, I strongly suggest including and testing open models as a backbone: this could help the research community to either address research gaps (if they don’t work properly) or use the proposed framework as a baseline for the future.

   - Metrics:
      - The AUC and MAD metrics use Gemini 1.5 pro to generate the synthetic data used to measure these scores. This means that the same model is used both in the framework to generate “feature descriptions” given a set of samples, and to generate “samples” given a feature description to evaluate them. This is a technical flaw that could bias the evaluation process and should be addressed. To solve this problem, a fair evaluation should use a family of models not used by any of the competitors in any of the steps.
     - Competitors and literature on the topic of automated interpretability for NLP and LLMs propose several metrics for measuring the quality of the generated explanations. For example, the Output-Centric method [24] discusses problems connected to methods using the highest activations and proposes some metrics to better evaluate them. In my opinion, testing the framework using a larger set of metrics could help the reader better understand the strengths and weaknesses of the proposed framework
      - The MAD metric seems to be associated with abnormally high variance in all the cases. Note that this behavior is different from what happens in the vision domain. Given the high variance associated with this metric, I have some reservations about its reliability to assess the quality of explanations in the NLP context.
  - Competitors: For each model tested, the framework is compared only against 1 competitor. This makes the current evaluation quite weak. The authors did a good job listing some of the alternative methods in section A.2. Including some of them in the comparison could strengthen the contribution of the paper (e.g., a useful baseline could be MaxAct)

- **Overgeneralization or missing context**:
    - From my understanding, the paper targets a specific niche: automated interpretability for explaining neurons of LLMs in the NLP domain. However, the narrative of the paper sometimes is more general than this niche and leads to overclaims, overgeneralizations, or in general could mislead practitioners. Some examples are reported below:
       - The abstract does not include any information about the context of LLMs and NLP, and some sentences reported in the abstract, like *"Unlike prior approaches that assign a single description per feature"* are definitely not true if applied to contexts beyond the specific niche targeted by the paper (e.g., there is a large body of literature on NLP and Vision that targets polysemanticity of neurons)
      - Line 31-32 This is an example of overgeneralization. Not all the existing methods provide a single description per feature. For example, several SAE methods, as the authors correctly mention in lines 28-30, try to study polysemanticity. The same holds for [14] when applied to NLP. If authors are referring (as I suspect) to a very specific niche and set of methods, this should be made clear in the text.

      To address this point, I would suggest revising and better positioning the abstract and the paper within the proper context and recognizing that the problem of polysemanticity has been studied for a long time outside this specific niche. Additionally, a discussion about why current and previous methods targeting polysemanticity cannot be applied (or produce inferior results) in this specific context would be appreciated and could improve the overall significance of the paper.

   - Human Interpretation: Quoting directly from the paper “Lastly, we took a step towards testing the alignment of our PRISM framework with human interpretation. [..] Our results highlight, that the PRISM framework not only provides multiple human interpretable descriptions for neurons but also aligns with the human interpretation of polysemanticity.” This sentence is based on the analysis/annotation process (section 5.2) performed by only one person: one of the co-authors. Therefore, the claim seems more general and there seems to be a mismatch between the claim and the experimental setup of the human interpretation. In this regard, I would suggest significantly reducing the claims about human interpretation or support them with more extensive analysis.

**Criteria under which my evaluation score could increase.**

Given the analysis above, unfortunately most of my concerns are related to the wording of the paper and it’s difficult to evaluate a change in the narrative without looking at the revised text (and NeurIPS doesn’t allow that). However, as a general guideline for the authors, and *assuming* that the other reviews don’t raise other major concerns that I may have overlooked:
1) Addressing the concern related to metrics (Adding more metrics and solving the problems related to the current metrics) and addressing the concern related to the specific configuration tested could increase the score up to 3 (borderline rejection).
2) Solving point 1 and adding more competitors (and thus a more extensive experimental setup) could increase score from 3 to 4 (borderline acceptance).
3) Addressing both the previous points and better aligning the wording and the narrative of the paper to the experimental setup would result in a score of 5

---

> ### Author Rebuttal · Authors · 2025-07-31
>
> We thank the reviewer for the detailed and constructive review and greatly appreciate the time and effort invested.
>
> **[W1] Experimental setup:**
>
> **[a] Method dependence on one specific LLM:**
> We understand the reviewer's concern about the potential dependency of our results on a specific configuration. To address the use of Gemini 1.5 Pro for description generation, we extended our evaluation on GPT-2 XL to include several open-source language models, namely Qwen3 32B, Phi-4, and DeepSeek R1, using the same setup as in our original experiments.
>
> As shown in the table below, Qwen3 32B achieves performance comparable to Gemini 1.5 Pro, demonstrating that our framework does not depend solely on a single model. While Phi-4 and DeepSeek R1 show slightly lower scores, they still follow the same qualitative trends, demonstrating that our method is robust and generalizes effectively across diverse language models, including open-source ones. We include this experiment and discussion in Section 4.2 (Sanity Checks).
>
> | Text Generator (description) | PRISM (max) |        | PRISM (mean) |         |
> |------------------------------|-------------|--------|--------------|---------|
> |                              | AUC (↑)     | MAD (↑)| AUC (↑)      | MAD (↑) |
> | Gemini 1.5 Pro (original) | 0.85 ± 0.25 | 2.98 ± 3.16 | 0.65 ± 0.36 | 1.24 ± 2.54 |
> | Qwen3 32B | 0.85 ± 0.24 | 2.63 ± 2.31 | 0.65 ± 0.36 | 1.20 ± 2.34 |
> | Phi-4 | 0.82 ± 0.26 | 2.18 ± 2.28 | 0.61 ± 0.38 | 0.83 ± 2.17 |
> | DeepSeek R1 | 0.79 ± 0.29 | 2.23 ± 2.32 | 0.61 ± 0.38 | 0.99 ± 2.33 |
>
> **[b1] Different LLMs for evaluation:**
> > "[...] a fair evaluation should use a family of models not used by any of the competitors in any of the steps."
>
> We agree that this is a relevant ablation. Thus, we used Qwen3 32B, Phi-4, and DeepSeek R1 to generate samples from the feature descriptions generated for GPT-2 XL. These models were not involved in the original generation step. As before, we computed AUC and MAD scores to assess the quality of the descriptions; results are shown in the table below for both PRISM and GPT-Explain.
>
> While overall scores are lower when Gemini 1.5 Pro is not used for evaluation, relative method rankings remain consistent. PRISM continues to outperform GPT-Explain, supporting the robustness of our evaluation setup and the generalizability of our framework. Along with [a], we include this experiment and discussion in Section 4.2 (Sanity Checks) and discuss the selection of different models for different steps (Section 3 and Discussion).
>
> | Text Generator (evaluation) | PRISM (max) |      | PRISM (mean) |         | GPT-Explain |    |
> |-----------------------------|-------------|------|--------------|---------|--------------|------|
> |                             | AUC (↑)     | MAD (↑)| AUC (↑)    | MAD (↑) | AUC (↑)    | MAD (↑)|
> | Gemini 1.5 Pro (original) | 0.85 ± 0.25 | 2.98 ± 3.16 | 0.65 ± 0.36 | 1.24 ± 2.54 | 0.64 ± 0.34 | 1.19 ± 2.38 |
> | Qwen3 32B | 0.58 ± 0.44 | 1.48 ± 3.62 | 0.53 ± 0.44 | 1.06 ± 3.56 | 0.54 ± 0.44 | 1.01 ± 3.52 |
> | Phi-4 | 0.61 ± 0.42 | 1.59 ± 3.32 | 0.54 ± 0.44 | 1.06 ± 3.28 | 0.56 ± 0.43 | 1.00 ± 3.20 |
> | DeepSeek R1 | 0.71 ± 0.35 | 2.02 ± 3.08 | 0.57 ± 0.42 | 1.08 ± 3.08 | 0.60 ± 0.40 | 1.22 ± 3.12 |
>
> **[b2] Different/output-centric Metrics for evaluation:**
> > "[...] testing the framework using a larger set of metrics [...]"
>
> We agree with the reviewer. To provide a more complete understanding of the strengths and weaknesses of our framework, we have incorporated a supporting evaluation using FADE [Pur25], which offers the following interpretability metrics:
>
> * *Activation-based metrics* based on the Absolute Gini Coefficient (Clarity and Responsiveness) and Absolute Precision  (Purity), which are closely related to our original AUC and MAD scores by quantifying concentration and discriminatory power of activations.
> * *An output-centric metric* (Faithfulness), which measures the causal influence of the discovered concept on model output, see also [24].
>
> Results are shown in the table below. PRISM (max) consistently outperforms GPT-Explain not only in semantic coherence (Clarity, Purity) but also in its impact on outputs (Faithfulness), further validating the method's effectiveness across a variety of interpretability criteria. As FADE's activation-based metrics align with ours, we followed the reviewer's suggestion and included FADE's output-centric faithfulness metric to Table 1, Section 4.3.
>
> |  GPT-2 XL | Clarity (↑) | Responsiveness (↑) | Purity (↑) | Faithfulness (↑) |
> | --- | --- | --- | --- | --- |
> | PRISM (max) |  0.681 ± 0.200 |	0.277 ± 0.146 	| 0.629 ± 0.130  	| 0.154 ± 0.090 |
> | PRISM (mean)|  0.405 ± 0.178 | 	0.151 ± 0.090 	| 0.416 ± 0.099 	| 0.134 ± 0.097  |
> | GPT-Explain |  0.513 ± 0.281 |	0.185 ± 0.175 	| 0.452 ± 0.177 	| 0.101 ± 0.072 |
>
> **[b3] Questionable reliability of MAD for NLP:**
> > "[...] I have some reservations about its reliability to assess the quality of explanations in the NLP context."
>
> In our paper, the MAD (Mean Activation Difference) metric serves as a *parametric* test as it quantifies the absolute difference in activation magnitudes between description-related examples and random inputs. Conceptually, the Input-based evaluation used by [24] in the context of NLP is closely related to our MAD metric. In [24], the authors test whether mean activations for the description-related examples exceed those of 'neutral' examples.
> As such parametric measures like MAD can be sensitive to outliers, particularly in NLP settings where activations are often characterized by heavy-tailed distributions. This is precisely why MAD is paired with the more robust AUC score, providing a fuller picture of neuron/feature behavior. We mention this limitation of MAD in Section 6 of the revised manuscript.
>
> **[c] Comparison to MaxAct Baseline:**
> We thank the reviewer for highlighting this. In response, we strengthened our experiments by comparing PRISM to MaxAct, a baseline that selects the five highest-activating sentences per feature, following [Bil23]. The table (due to character limits, the table appears in our **response to Reviewer 2jDP**, under **[W2] Percentile Sampling**) shows PRISM (max) outperforms MaxAct across most metrics and models, especially in AUC. We include MaxAct as a baseline for all models in Table 1, Section 4.3 of the revised manuscript.
>
> We agree that broader comparisons are valuable. However, not all methods listed in Appendix A.2 have available feature descriptions, which limits reproducibility. We prioritized baselines with publicly accessible feature descriptions.
>
> **[W2] Overgeneralization or missing context:**
>
> **[a] Address Niche:**
> We thank the reviewer for pointing out where the narrative of the paper could lead to overgeneralizations. Our work indeed targets a specific niche of methods that assign explanations to individual features, like standard neurons or SAE-discovered features, within LLMs for NLP, and does not cover polysemanticity analyses across domains. We focus on improving feature-description methods for LLMs that typically provide a single textual explanation per neuron/feature, by instead assigning multiple explanations, potentially capturing the diverse behaviors of a single feature. We recognize that:
> - Many existing SAE-based methods explicitly aim to reduce polysemanticity by producing sparse and more monosemantic features.
> - There is a substantial body of prior work (e.g., in both NLP and Vision) analyzing polysemantic neurons.
>
> To address scope ambiguities in the original manuscript, we have revised the text to clearly articulate our focus throughout. Specifically we rewrite:
>
> - abstract lines 2-5 to:
> > "Within the context of large language models (LLMs) for natural language processing (NLP), current automated neuron-level feature description methods face two key challenges: limited robustness and the assumption that each neuron encodes a single concept (monosemanticity), despite increasing evidence of polysemanticity."
>
> -  abstract lines 9–11 to:
> > "[...] framework specifically designed to capture the complexity of features in LLMs. Unlike approaches that assign a single description per neuron, common in many automated interpretability methods in NLP, PRISM produces more nuanced descriptions that account for both monosemantic and polysemantic behavior."
>
> - lines 31–32 to:
> > "While the problem of polysemanticity is generally addressed through the extraction of sparse features, such as via sparse autoencoders, feature description methods, which aim to explain the functional purpose of individual features, typically provide a single explanation per feature [20, 21, 22, 23, 24]. This can limit the ability to capture the full range of patterns a feature may represent."
>
> **[b] Human Interpretation:**
> We expanded the human evaluation in Section 5.2 by involving seven additional non-author participants, results show good alignment between human judgments of polysemanticity and the PRISM metric; due to character constraints further details can be found in our **response to Reviewer 2jDP** under **[W5] Human evaluation**.
>
> **[Q1] Question:**
> Techniques like SAEs seek to promote disentanglement but often exhibit high reconstruction errors, model faithfulness and lack theoretical guarantees that the learned features are actually monosemantic (see [Shar25], Sec. 2.1). As a result, individual features may still respond to multiple concepts, which limits interpretability. We now refer the reader to [Shar25] for a more comprehensive discussion of SAEs and their limitations
>
> Please let us know if any further comments or questions arise. We are happy to address them during the discussion period.
>
> **Refs**
> - [Pur25] Puri et al., FADE: Why Bad Descriptions Happen to Good Features. ACL 2025.
> - [Bil23] Bills et al., Language models can explain neurons in language models. OpenAI 2023.
> - [Shar25] Sharkey et al., Open Problems in Mechanistic Interpretability. Arxiv 2025.

---

> ### Comment · Reviewer_9BmE · 2025-08-03
>
> Thank you to the authors for acknowledging the limitations of the previous version and for their answer. The comparison with competitors is still limited, as acknowledged by the authors, but overall I believe the new evidence better supports the claims and I will raise my scores according to the policy I stated above and the overall discussion.
>
> I would like to raise a couple of follow-up questions based on the rebuttal:
> - Over how many runs are the new results computed? The variance across all the metrics seems quite high
> - I raised this point earlier, but I may have missed the response: why can’t existing methods targeting polysemanticity be applied (or why do they perform poorly) in the specific context of this paper? This point is related to the contextualization of the paper with respect to the literature studying polysemanticity.

---

> > ### Author Response · Authors · 2025-08-05
> >
> > We sincerely thank the reviewer for their feedback and for updating their score; it is greatly appreciated!
> >
> > > 1) Over how many runs are the new results computed? The variance across all the metrics seems quite high
> >
> > For the computation of the FADE metrics, we followed the same experimental setup as in the main paper, using the same number of samples, feature description clusters, and layers. For our ablation studies, we modified only the components explicitly mentioned, keeping all other settings identical to those in the main paper.
> > Regarding the robustness of evaluation scores, this is a well-known challenge in the context of feature descriptions. The authors of FADE [Pur25] report similarly high variance, particularly when evaluating on a random subset of neurons, as in our setup, rather than selecting a subset of top-performing neurons (see Fig. 3 in [Pur25]).  Similarly, [Bil23] report that only a small subset of neurons (1.7\%) achieve above 0.7 on their explanation score metric, which measures the correlation between simulated and actual neuron behavior. Despite this variance, we observe that the relative trends across methods remain stable, supporting the reliability of our conclusions.
> >
> >
> > > 2) I raised this point earlier, but I may have missed the response: why can’t existing methods targeting polysemanticity be applied (or why do they perform poorly) in the specific context of this paper? This point is related to the contextualization of the paper with respect to the literature studying polysemanticity.
> >
> > Thank you for raising this important point. We realize that the distinction between concept extraction and concept description may not have been sufficiently clear in the original version of the manuscript, which could lead to misunderstandings about the scope of our work. We have revised the text to clarify that our research specifically focuses on concept description methods, rather than concept extraction approaches such as SAEs. To avoid further confusion, we now clearly distinguish between these two categories, emphasizing that our contribution centers on description rather than extraction.
> >
> > Our paper acknowledges existing methods like SAEs to target polysemanticity, and we note that SAEs perform favorably in our evaluation, particularly with Gemma Scope showing the highest monosemanticity across layers compared to non-SAE models.
> >
> > As our study does not aim to make prescriptive recommendations about the use of SAEs, we have revised the relevant sections of the manuscript (e.g., lines 30 and 196–197) to avoid drawing conclusions about potential limitations of SAEs.
> >
> >
> > **Refs**
> >
> > - [Pur25] Puri et al., FADE: Why Bad Descriptions Happen to Good Features. ACL 2025.
> > - [Bil23] Bills et al., Language models can explain neurons in language models. OpenAI 2023.

---

> > > ### Comment · Reviewer_9BmE · 2025-08-06
> > >
> > > Thanks to the authors for the additional clarifications. I have no further questions and will raise my score in accordance with the policy described in my initial review.
> > >
> > > Overall, the new experimental evidence better supports the paper's claims. There are still some weaknesses related to the limited comparison against other concept description methods and the small user study (only 7 participants), which make the paper borderline. Nevertheless, my final recommendation will reflect the improvements provided during the rebuttal phase and will lean towards acceptance.

---

> > > > ### Author Response · Authors · 2025-08-08
> > > >
> > > > Thank you again for your constructive review. The expanded benchmarks and user pilot study helped clarify the value of PRISM. We agree that a broader user study on polysemanticity and LLMs presents an exciting and timely direction for future work, and we appreciate your final recommendation leaning toward acceptance.

---

### Official Review · Reviewer_2jDP · 2025-07-01

**Clarity:** 1
**Significance:** 2
**Originality:** 3
**Rating:** 4
**Confidence:** 2

**Summary:**

This paper proposed a multi-concept feature description framework called PRISM, which applies to the description of both polysemantic and monosemantic features. The proposed framework reveals that Individual features encode a highly diverse and heterogeneous set of concepts. The framework also shows a high alignment with the human interpretation of polysemanticity.

**Questions:**

See Weaknesses.

**Ethical Concerns:**

["NO or VERY MINOR ethics concerns only"]

**Final Justification:**

The rebuttal solved my concerns. I update my scores to lean towards borderline acceptance. I cannot give a higher score because, although overall effective, this method carries the risk of introducing unfaithful descriptions.

**Limitations:**

NA.

**Paper Formatting Concerns:**

NA.

**Quality:**

2

**Strengths And Weaknesses:**

**Strengths.**

1. The proposed framework allows the description of polysemantic features, and the method is intuitive.
2. Alignment with the human interpretation of polysemanticity is tested.

**Weaknesses.**

1. The paper is poorly written and hard to read. Specifically,
- In Eq (3), $s_j$ represents a concise natural language summary, but it represents sentence embedding in Eq (4).
- Section 3.2, the exact descriptions of AUC and MAD are repeated and shown twice.
- Section 3.2. What are the target concept and control data points? They are not introduced in the main paper. How is PRISM evaluated through AUC and MAD? Is the target concept the generated cluster labels? Why do higher AUC and MAD indicate better description? Also, I think Figure 6 in Appendix A.3 should be put in Section 3.2 for better understanding.
2. In section 3.1, why do we need a grid of high-percentile levels in percentile sampling, instead of using only the top activations?
3. Line 148. How is the number of clusters determined?
4. Line 154. Why are only positive activations considered? Is it possible that a sample has a large negative activation?
5. Section 5.2 includes only one human tester, which may lead to great bias.

---

> ### Author Rebuttal · Authors · 2025-07-31
>
> We thank the reviewer for the thoughtful comments. We regret the clarity fell short for this reviewer, we appreciate the chance to improve. While other reviewers found the writing and contributions clear, we recognize that presentation can be refined to suit a broader audience, and we have worked carefully to address these concerns.
>
> **[W1] Writing:**
> > "The paper is poorly written and hard to read. Specifically, [...]" (a-e)
>
> Following the reviewer's comments, we have revised relevant sections to improve clarity and readability. While we are unable to submit a revised manuscript at this stage, we hope our clarifications below help address any potential for confusion.
>
> **[a] Equations 3 and 4:** We thank the reviewer for pointing out that the symbol $s_j$ is overloaded in both equations. We disambiguate this by using distinct symbols $s_j$ for the natural language summary and $\tau_j = e(s_j), \tau_j \in \mathbb{R}^T$ for the embeddings of a specific natural language summary. We correct Eq. 4. accordingly:
> > $\text{cos}(\theta) = \frac{\sum_t^T \tau_{i,t}\tau_{j,t}}{\sqrt{\sum_t^T \tau_{i,t}^2} \sqrt{\sum_t^T \tau_{j,t}^2}}$.
>
> **[b] Duplicate description:**
> We removed the unintentional repetition.
>
> **[c] Definition of target concept and control data points:**
> The target concept dataset consists of sentences generated based on cluster labels for a given neuron. Control data consists of random sentences which are not associated with the concept.
> We previously defined this in Appendix A.3, but based on the reviewer's helpful remark, we have now moved this explanation to the main text (Section 3.2) to improve clarity and ensure the evaluation setup is more easily understood.
>
> **[d] AUC and MAD as evaluation metric:**
> AUC and MAD scores are used to quantify whether example sentences generated based on each cluster label activate the corresponding neuron more strongly than the random sentences in the control dataset (see answer [W1][c]). Higher AUC and lower MAD indicate more semantically aligned and distinctive concept labels for the different cluster.
>
> **[e] Placement of Figure 6 (Appendix A.3):**  We agree that this figure is informative and helps illustrate the evaluation process. However, due to the length restrictions of 9 pages, we previously struggled to integrate the Figure in the main body. Given the opportunity we will aim to move it into the main paper for better clarity.
>
> **[W2] Percentile Sampling:**
> > "In section 3.1, why do we need a grid of high-percentile levels in percentile sampling, instead of using only the top activations?"
>
> We use a grid of high-percentile levels rather than a fixed top-k cutoff to retain diverse activation contexts that could be excluded by selecting only the top-k samples. Absolute activation can vary strongly across neurons, layers and even corpora, so using a fixed cut‑off (e.g. the top‑k sentences) risks discarding genuine but moderately‑scaled patterns. Instead, sampling sentences uniformly from the 99th to 100th percentile with a step size of 1e-5 retains focus on strong activations while enabling broader coverage of distinct patterns that activate the same feature. As demonstrated in our experiments, this signal allows capturing mutli-concept feature descriptions for neurons that have previously been assigned a single description.
>
> We empirically evaluated this design choice by comparing PRISM to a baseline approach (MaxAct) that selects the five highest-activating sentences per feature, following [Bil23], using a subset of the C4 dataset. PRISM (max) outperforms MaxAct on most metrics and models, particularly in AUC (see table below). This supports the benefit of our percentile-based sampling for interpretability and concept coverage.
> | Model | Method | AUC (↑) | MAD (↑) |
> |--------|-------|---------|---------|
> | GPT-2 XL | PRISM (max) | 0.85 ± 0.25 | 2.98 ± 3.16 |
> | GPT-2 XL | PRISM (mean) | 0.65 ± 0.36 | 1.24 ± 2.54 |
> | GPT-2 XL | GPT-Explain | 0.64 ± 0.34 | 1.19 ± 2.38 |
> | GPT-2 XL | MaxAct* | 0.63 ± 0.40 | 1.82 ± 3.76 |
> | Gemma Scope | PRISM (max) | 0.54 ± 0.33 | 5.26 ± 18.51 |
> | Gemma Scope | PRISM (mean) | 0.43 ± 0.31 | 2.98 ± 14.68 |
> | Gemma Scope | Output-Centric | 0.58 ± 0.35 | 12.24 ± 40.16 |
> | Gemma Scope | MaxAct* | 0.53 ± 0.31 | 5.35 ± 28.44 |
>
> **[W3] Number of Clusters:**
> > "How is the number of clusters determined?"
>
> We thank the reviewer for raising this important question. In our original implementation, we chose 5 clusters as a default, as this provided a balance between granularity and interpretability, allowing for multiple semantic patterns to emerge while limiting potential redundancy or incoherence. We analyze the effect of varying the number of clusters on PRISM's performance across multiple values of k and report both max and mean cluster scores across features in GPT-2 XL, using AUC and MAD as in our main evaluation (see table below). Increasing the number of clusters raises PRISM max scores by allowing clusters to capture more specific activation patterns, improving labeling of the most coherent ones, but lowers the PRISM mean score as coherent patterns are split into statistically indistinguishable subclusters. We have added clarification regarding the choice of the number of clusters to Section 4.1 and discuss implications of increasing/decreasing k in our discussion in Section 6.
> | k Clusters | PRISM (max) |        | PRISM (mean) |         |
> |------------|-------------|--------|--------------|---------|
> |            | AUC (↑)     | MAD (↑)| AUC (↑)      | MAD (↑) |
> | 5 (original) | 0.85 ± 0.25 | 2.98 ± 3.16 | 0.65 ± 0.36 | 1.24 ± 2.54 |
> | 1 | 0.75 ± 0.31 | 1.79 ± 2.35 | 0.75 ± 0.31 | 1.79 ± 2.37 |
> | 3 | 0.82 ± 0.28 | 2.75 ± 2.69 | 0.69 ± 0.35 | 1.52 ± 2.45 |
> | 10 | 0.88 ± 0.19 | 3.05 ± 2.62 | 0.61 ± 0.37 | 0.99 ± 2.24 |
>
> **[W4] Positive/negative activations:**
> > "Why are only positive activations considered? Is it possible that a sample has a large negative activation?"
>
> We focus on positive activations as we are interested in characterizing features when they are strongly activated ("fire") in response to a specific input.
> While negative activations can occur, their interpretation remains unclear and is still debated (see [Mol20]).
> Given that common activation functions in state-of-the-art LLMs, such as ReLU and GeLU, retain only positive or weakly negative signals, we follow prior work on feature description methods (e.g., [Her22, Bil23]) and threshold activations to retain only the top-activating samples. That said, our framework is compatible with extracting descriptions across the full activation range.
>
> **[W5] Human evaluation:** Inspired by the reviewer's comments, we decided to extend the scope of the human evaluation in Section 5.2 by conducting a larger human evaluation with seven participants, all of whom are non-authors. Participants were recruited from our institutions and compensated. We provide additional details on participant recruitment, task instructions, and the study setup in the Appendix of the revised manuscript for full transparency.
>
> Each participant was presented with 8 groups of sentence clusters, where each group corresponded to one feature and consisted of 5 clusters of highly activating sentences (20 sentences per cluster), along with highlighted tokens. For each cluster, participants were tasked to write a short textual description, resulting in 5 human-generated descriptions per feature. They then rated the pairwise similarity between these descriptions on an 11-point scale (0–10), where 10 indicates very high similarity. We report the average of these ratings as the Human Score. In addition, we compute Human Cosine Similarity using the same embedding model and method used for computing the PRISM polysemanticity score, with the only change being the use of human-generated descriptions instead of model-generated ones.
>
> The table below presents the results, sorted by PRISM score (lower values indicate higher polysemanticity). As expected, features with low PRISM scores receive lower human similarity ratings and lower embedding-based similarity. For example, the feature from GPT-2 Small (layer 5, feature 30319) has a low PRISM score (0.30), and is judged by participants to be semantically diverse (Human Score: 0.40). In contrast, features with high PRISM scores, such as those from Gemma Scope (features 10531 and 603), show strong human agreement (Human Score: 0.90 and 0.80, respectively) and high cosine similarity (0.79 and 0.75). These findings provide both qualitative and quantitative support for the PRISM metric, showing that it aligns well with the human interpretation polysemanticity. We have replaced the original human evaluation with these new, extended results and discussion in Section 5.2 of the revised manuscript.
> | Model | Layer | Feature | Human Score | Human Cos. Sim. | PRISM Score |
> | --- | --- | --- | --- | --- | --- |
> | GPT-2 Small | 5 | 30319 | 0.40 ± 0.29 | 0.44 ± 0.14 | 0.30 ± 0.17 |
> | GPT-2 XL | 40 | 6067 | 0.40 ± 0.14 | 0.46 ± 0.10 | 0.38 ± 0.07 |
> | Llama 3.1 8B Instruct | 0 | 12733 | 0.50 ± 0.17 | 0.52 ± 0.16 | 0.41 ± 0.08 |
> | GPT-2 XL | 20 | 3313 | 0.70 ± 0.10 | 0.53 ± 0.15 | 0.59 ± 0.16 |
> | GPT-2 Small | 10 | 9896 | 0.80 ± 0.17 | 0.69 ± 0.19 | 0.60 ± 0.09 |
> | Llama 3.1 8B Instruct | 0 | 10095 | 0.70 ± 0.21 | 0.59 ± 0.10 | 0.65 ± 0.06 |
> | Gemma Scope | 0 | 10531 | 0.90 ± 0.16 | 0.79 ± 0.23 | 0.68 ± 0.03 |
> | Gemma Scope | 10 | 603 | 0.80 ± 0.17 | 0.75 ± 0.25 | 0.74 ± 0.08 |
>
> We hope our responses and changes have effectively addressed the concerns. If any further questions or suggestions arise, we are happy to continue the discussion.
>
> **Refs**
> - [Mol20] C Molnar. Interpretable Machine Learning Book,
> Chapter 27. 2020.
> - [Her22] Hernandez et al., Natural language descriptions of deep visual features. ICLR 2022.
> - [Bil23] Bills et al., Language models can explain neurons in language models. OpenAI 2023.

---

> > ### Comment · Reviewer_2jDP · 2025-08-04
> >
> > Thank you for the rebuttal. The explanations and additional experiments solved most of my concerns.
> >
> > I still have one additional concern. Compared with PRISM (max), the performance of PRISM (mean) drops severely, and the performance gap becomes even larger as $k$ increases. I wonder how do the less matching descriptions perform, e.g., the performance for PRISM (min). Do they still activate the corresponding neuron significantly more strongly than random sentences? If not, these less matching descriptions should not count when considering the polysemanticity of a neuron.

---

> > > ### Author Response · Authors · 2025-08-05
> > >
> > > We thank the reviewer for the follow-up question. It is correct that PRISM (mean) and PRISM (min) show lower scores compared to PRISM (max). We report the minimum PRISM score for GPT-2 XL in the table below.
> > >
> > > | Model | Method | AUC (↑) | MAD (↑) |
> > > |--------|-------|---------|---------|
> > > | GPT-2 XL | PRISM (max) | 0.85 ± 0.25 | 2.98 ± 3.16 |
> > > | GPT-2 XL | PRISM (mean) | 0.65 ± 0.36 | 1.24 ± 2.54 |
> > > | GPT-2 XL | PRISM (min) | 0.48 ± 0.38 | 0.04 ± 1.80 |
> > >
> > > The low PRISM (min) and PRISM (mean) scores are expected given our design: we assign 5 clusters to all neurons, regardless of whether they are mono- or polysemantic and how many concepts activate the given neuron, to systematically assess potential polysemanticity. For both mono- or polysemantic neurons, this often results in some clusters being less meaningful which naturally lowers the mean and minimum scores. For example, in layer $0$, neuron $5551$, the lowest scoring descriptions among the $5$ cluster descriptions is "Food, tools/devices and cosmetics/accessories", with an AUC of $0.024$ (minimum PRISM score). In contrast, for neuron $6067$ in layer $40$, the worst performing description is "Events, often with a time and/or a date", but the minimum PRISM score is much higher at $0.838$. This is why we deliberately report all descriptions along with the associated AUC and MAD values, allowing users to interpret the full distribution of feature descriptions. Our goal is to avoid injecting additional assumptions or thresholds that might obscure or pre-filter cases of potential polysemanticity. While PRISM (min) descriptions may not strongly activate the neuron above random, their inclusion supports a more transparent and assumption-free characterization across the neuron population. To improve the reader's understanding of the methods variation we have added additional examples in the Appendix of the current manuscript version and extended the discussion in future work to include potential filtering of descriptions based on their AUC PRISM score (in the calculation of the polysemanticity score).

---

> > > > ### Comment · Reviewer_2jDP · 2025-08-07
> > > >
> > > > Thank you for the reply. I have no further questions and I will update my rating accordingly.

---

### Official Review · Reviewer_MiB7 · 2025-07-02

**Clarity:** 3
**Significance:** 3
**Originality:** 3
**Rating:** 5
**Confidence:** 3

**Summary:**

The paper presents PRISM: a mechanistic interpretability technique to assign labels to individual language model features. PRISM's primary novelty is in its ability to discern polysemantic features in language models.

For each feature, PRISM selects input sentences from high-percentiles of activations for the feature being studied. These sentences are then mapped to real feature vectors, using a text embedding model based on Qwen. These feature vectors are then clustered using $k$-means clustering with a fixed number of clusters. An LLM is then given the top few sentences from each cluster to produce human readable feature labels.  The explanations provided by the model are evaluated using CoSy, and ROC-AUC and MAD scores are reported.

**Questions:**

- What are the failure modes of smaller open source LLMs when used to generate the human readable descriptions?
- What alternatives may be employed to proprietary LLMS to generate descriptions?

**Ethical Concerns:**

["NO or VERY MINOR ethics concerns only"]

**Final Justification:**

The author's responses were informative, and I believe that this is a strong paper.

**Limitations:**

yes

**Paper Formatting Concerns:**

no concerns

**Quality:**

4

**Strengths And Weaknesses:**

Strengths:
- The paper proposes a novel method to elicit polysemantic features from language models
- Very thorough experimental evaluation of the proposed method is performed
- The method is explained clearly

Weaknesses:
- A key weakness of the method is that high capacity (even proprietary) LLMs are required to produce high-quality descriptions of features.  The authors acknowledge this in footnote 7 and the limitations section. It is at first glance surprising to me that the simple task of assigning a human readable label to a group of sentences is beyond the reach of standard open-source LLMs. I would encourage the authors to suggest alternative lower-cost approaches to circumvent this issue.
- The sentence in line 174 is difficult for me to parse. Which of the two scores is the random baseline, and which is PRISM?

---

> ### Author Rebuttal · Authors · 2025-07-31
>
> We sincerely thank the reviewer for their thoughtful, detailed, and constructive feedback. We are especially grateful for the reviewer's appreciation of the paper's clarity, the novelty and simplicity of PRISM, and the extent of our experimental evaluation. The reviewer's comments have helped us both sharpen the presentation and deepen our analysis. We have carefully considered all concerns and made substantial efforts to address each point. Below, we respond to the main weaknesses and questions raised. We make sure the revised version reflects these improvements and communicates the work more effectively. Please let us know if any further concerns remain.
>
> **[W1/Q2] Open Source LLMs:**
>
> > "[...] high capacity (even proprietary) LLMs are required to produce high-quality descriptions of features."
>
> > "What alternatives may be employed to proprietary LLMS to generate descriptions?"
>
> We appreciate the reviewer's concern regarding whether high-capacity or proprietary LLMs are necessary in the PRISM framework to produce high-quality feature descriptions. To address this concern, we extended our evaluation beyond Gemini 1.5 Pro by benchmarking several open-source alternatives: Qwen3 32B, Phi-4, and DeepSeek R1. These models were used to generate feature descriptions for GPT-2 XL using the same procedure as in our original setup. As shown in the table below, Qwen3 32B achieves performance comparable to Gemini 1.5 Pro, demonstrating that our framework does not depend solely on a single model. While Phi-4 and DeepSeek R1 show slightly lower scores, they still follow the same qualitative trends, demonstrating that our method is robust and generalizes effectively across a range of language models, including accessible open-source options. We have included this experiment and discussion to Section 4.2 (Sanity Checks) of the revised manuscript.
>
> | Text Generator (description) | PRISM (max) |        | PRISM (mean) |         |
> |------------------------------|-------------|--------|--------------|---------|
> |                              | AUC (↑)     | MAD (↑)| AUC (↑)      | MAD (↑) |
> | Gemini 1.5 Pro (original) | 0.85 ± 0.25 | 2.98 ± 3.16 | 0.65 ± 0.36 | 1.24 ± 2.54 |
> | Qwen3 32B | 0.85 ± 0.24 | 2.63 ± 2.31 | 0.65 ± 0.36 | 1.20 ± 2.34 |
> | Phi-4 | 0.82 ± 0.26 | 2.18 ± 2.28 | 0.61 ± 0.38 | 0.83 ± 2.17 |
> | DeepSeek R1 | 0.79 ± 0.29 | 2.23 ± 2.32 | 0.61 ± 0.38 | 0.99 ± 2.33 |
>
> **[W2] Unclear sentence in line 174:**
>
> > "Which of the two scores is the random baseline, and which is PRISM?"
>
> We understand the confusion regarding the meaning of this sentence. To clarify, the AUC of 0.85 ± 0.25 and MAD of 2.98 ± 3.16 refer to the PRISM baseline (with descriptions generated from semantically meaningful clusters), while the AUC of 0.65 ± 0.35 and MAD of 1.35 ± 2.36 correspond to the randomized control (with cluster descriptions shuffled across features).
>
> We have rephrased the sentence and included the results in tabular form for GPT-2 XL in the revised manuscript to make this distinction clearer. The updated sentence now reads:
>
> > "The lower score for PRISM in the random scenario (Random Descriptions) compared to PRISM without randomization (Baseline) indicates that the PRISM descriptions truly depend on the content of the clusters because randomizing them leads to a decrease in performance."
>
> The corresponding results are now summarized in the following table:
>
> |            | PRISM (max) |        | PRISM (mean) |         |
> |------------|-------------|--------|--------------|---------|
> |            | AUC (↑)     | MAD (↑)| AUC (↑)      | MAD (↑) |
> | Baseline   | 0.85 ± 0.25 | 2.98 ± 3.16 | 0.65 ± 0.36 | 1.24 ± 2.54 |
> | Random Descriptions | 0.65 ± 0.35 | 1.35 ± 2.36 | 0.52 ± 0.40 | 0.52 ± 2.19 |
>
> **[Q1] Smaller Open Source LLMs:**
>
> > "What are the failure modes of smaller open source LLMs when used to generate the human readable descriptions?"
>
> In an effort to characterize failure modes, including smaller open-source models, we provide the evaluation score in the table in [W1/Q2]. While the decrease in description quality measured by AUC and MAD supports prior findings in the literature on the limitations of smaller LLMs [Pur25], score decreases are small indicating PRISM's robustness towards the LLM choice.
>
> In our analysis of the lighter LLMs (e.g. Qwen2.5 7B Instruct, Llama 3.1 8B Instruct), we found that these models are limited due to a combination of issues: they tend to over-generalize, produce overly verbose or vague outputs, struggle with instruction adherence, and in some cases, ignore instructions altogether. These shortcomings reduce the consistency and specificity of the generated descriptions. As noted in [W1/Q2], we have incorporated these findings into the revised manuscript along with an expanded discussion of these limitations and supporting results.
>
> **Refs**
> - [Pur25] Puri et al., FADE: Why Bad Descriptions Happen to Good Features. ACL 2025.

---

> > ### Comment · Reviewer_MiB7 · 2025-08-05
> >
> > Thank you for your detailed response. The method presented is novel and useful.
> > I believe that it should be included at NeurIPS, and my review and score reflect the same.

---

### Official Review · Reviewer_axts · 2025-07-02

**Clarity:** 3
**Significance:** 3
**Originality:** 3
**Rating:** 4
**Confidence:** 4

**Summary:**

The paper proposes a novel method PRISM, to generate automated explanations for arbitrary features or neurons in a LLM that respects polysemanticity. The method functions by performing percentile sampling for activation of any given feature/neuron to select top activating samples but with more variety than doing direct top-k sampling. It then clusters these samples based on their semantics and for each cluster generates a summary using a language model using the most activating members of the cluster. The method is tested on various language models and also demonstrated on raw neurons and directional activations.

**Questions:**

1. Could you address weakness 1.

2. Weakness 2.

3. Weakness 3

4. Is the method robust for the choice of $e$ (line 100), and the sentence embedding model (line 108)?

5. In practice, did you observe biases in your descriptions because of the choice of LLM (for summary generation) or sentence embedding function?

6. Looking at the qualitative examples, my understanding of what the paper considers as distinction between monosemantic and polysemantic concepts is less clear. If you can assign meaningful metalabels for features, are the underlying features really polysemantic? The metalabel acts in this case as the semantic captured by the feature which should thus be considered monosemantic. For eg. If a feature activates for names of animals then having two different clusters, one for dog and one for cat, should not mean its polysemantic. It's good you have polysemanticity score to quantify this but I am getting confused by the qualitative examples. Most of them I wouldn't consider polysemantic. The only exception is the Fig. 5 top where also I only see two different semantics (Occasions/festivities and encryption/rules etc.). Could you make your stand clear in general and specify it in paper for all the qualitative examples? Also maybe visualize more polysemantic concepts in the main paper as its the core point.

I like the paper but there are some glaring gaps that need to be filled.

**Ethical Concerns:**

["NO or VERY MINOR ethics concerns only"]

**Final Justification:**

The authors responded with clarity and transparency to my questions. Most of my concerns were addressed completely. I see two limitations that were came up in discussion with me or other reviewers:

(1) In a small subset of cases the authors do observe neurons where their sampling strategy could benefit from updates by taking into account relative activations for a given neuron, but considering all the arguments, I would still agree the current proposed method is useful and has been shown to be effective.

(2) The relatively limited size of subjective evaluation as noted by Reviewer 9BmE.

Overall, in view of the interesting problem, novel proposal, and clear and comprehensive experiments, the positives outweight the limitations for me. I lean towards acceptance of this paper.

**Limitations:**

The authors have stated reasonable limitations although I believe there are some more limitations (mentioned in the review)

**Quality:**

2

**Strengths And Weaknesses:**

Strengths:
- The writing and paper presentation is very clear in most places.
- I find the problem statement of generating automated explanations to capture polysemanticity interesting and relevant.
- The proposed solution is simple and novel.
- Experimental evaluation is extensive in terms of multiple models and different types of experiments to analyze the approach.


Weaknesses:
- W1: The method does not take into account the activation of the individual points relative to the highest activation. For instance, if the 99th percentile input has activation an order of magnitude lower than 100th percentile, it should not be considered for clustering.
- W2: While the AUC and MAD metrics quantify if descriptions correspond to input samples where a feature/neuron activates, I didn't find any evaluation regarding "how" much of the feature is "explained" through the obtained descriptions. In other words, are the descriptions of $k=5$ clusters enough to "cover" all samples that can activate the given feature? This evaluation should strengthen the paper and I would expect favourable results for this method compared to baselines since it tries to capture polysemanticity. Maybe it also provides a handle to better choose hyper parameters for generating descriptions for a feature.
- W3: Sec 5.2 experiment is weak. It should be repeated with multiple annotators not just one. Even though it might have positive indications, it's not sound at the moment.

(Minor corrections)
- eq. 3 -> $C_k$ should be $C_j$
- $s_j$ is used for both natural language summary in eq. 3 as well as the embedding notation in eq. 4.
- You should explain the high variance in MAD scores better and directly in the main paper (Appendix Fig. 10 I suppose is for that)

---

> ### Author Rebuttal · Authors · 2025-07-31
>
> We would like to thank the reviewer for the detailed feedback and helpful suggestions. We have incorporated the suggested minor corrections into the revised manuscript and address the comments below.
>
> **[W1/Q1] Percentile Sampling:**
>
> > "The method does not take into account the activation of the individual points relative to the highest activation. For instance, if the 99th percentile input has activation an order of magnitude lower than 100th percentile, it should not be considered for clustering."
>
> Absolute activation magnitudes vary substantially across neurons, layers, and even datasets. Relying on a fixed threshold (e.g., selecting only the top-k sentences) risks excluding informative patterns with moderate but consistent activations. To address this, our method samples sentences uniformly from the 99th to 100th percentile with a step size of 1e-5, allowing us to focus exclusively on the upper tail of the activation distribution while preserving semantic diversity.
>
> This percentile-based sampling is a core distinction between PRISM and other feature description methods that are typically based on top-k approaches.
> We demonstrate (e.g., Table 2 for quantitative results and Figure 4 for qualitative results) that sampling from the 99th percentile already yields sentences that both robustly activate the feature and reflect semantic diversity, supporting a more holistic understanding of neuron behavior.
>
> **[W2/Q2] Setting of Cluster Size:**
>
> > "While the AUC and MAD metrics quantify if descriptions correspond to input samples where a feature/neuron activates, I didn't find any evaluation regarding "how" much of the feature is "explained" through the obtained descriptions. In other words, are the descriptions of clusters enough to "cover" all samples that can activate the given feature?"
>
> We would like to clarify that the AUC and MAD metrics are designed to quantify how strongly a neuron/feature reacts to inputs corresponding to a given description, in comparison to a random input set. As described in Appendix A.3, we evaluate a proposed description (e.g., “Quantities, specifically numbers, and time periods”) by prompting an LLM to generate related sentences (see Figure 7 in Appendix for exact prompt details). We then compare the activations of the neuron/feature on these description-related sentences against its activations on random inputs. The separation between the two distributions is measured using a nonparametric metric (AUC, where 1.0 is ideal) and a parametric one (MAD, where higher is better). A high score indicates that the neuron/feature responds significantly more to inputs aligned with the description, which suggests a faithful explanation.
>
> >  "In other words, are the descriptions of clusters enough to "cover" all samples that can activate the given feature?"
>
> This is indeed a limitation of all current feature description methods, as these depend on a reference dataset, limiting the descriptions to cover concepts that occur in a subset of these samples.
>
> Our approach progresses on the challenge of finding richer, multi-faceted descriptions of each feature, describing it with a set of several distinct concepts (in comparison to a single description from standard approaches). However, like all such methods, PRISM cannot guarantee that the provided descriptions are the only or most salient concepts for a feature; rather, they represent key concepts the feature reliably responds to. We have now extended our brief discussion of this limitation in Section 6 of the revised manuscript.
>
> To further examine this, we analyzed PRISM's performance across different numbers of clusters (k) for generating feature descriptions for GPT-2 XL. As shown in the table below, increasing k improves the best-case description quality, as reflected in higher PRISM max scores, but tends to reduce the average interpretability across all clusters (lower PRISM mean scores). This highlights a tradeoff: more clusters can yield more precise and targeted descriptions, but may also introduce redundancy or semantic overlap. The results agree with our intuition that with larger numbers of clusters, the best label becomes more precise (PRISM max score) while the overall score (PRISM mean score) decreases since now monosemantic labels are over-labeled. We have added this experiment and discussion to the Appendix of the revised manuscript.
>
>
> | k Clusters | PRISM (max) |        | PRISM (mean) |         |
> |------------|-------------|--------|--------------|---------|
> |            | AUC (↑)     | MAD (↑)| AUC (↑)      | MAD (↑) |
> | 5 (original) | 0.85 ± 0.25 | 2.98 ± 3.16 | 0.65 ± 0.36 | 1.24 ± 2.54 |
> | 1 | 0.75 ± 0.31 | 1.79 ± 2.35 | 0.75 ± 0.31 | 1.79 ± 2.37 |
> | 3 | 0.82 ± 0.28 | 2.75 ± 2.69 | 0.69 ± 0.35 | 1.52 ± 2.45 |
> | 10 | 0.88 ± 0.19 | 3.05 ± 2.62 | 0.61 ± 0.37 | 0.99 ± 2.24 |
>
> **[W3/Q3] Human Evaluation:**
>
> > "[Experiment in Section 5.2] should be repeated with multiple annotators"
>
> We expanded the human evaluation in Section 5.2 by involving seven additional non-author participants, results show good alignment between human judgments of polysemanticity and the PRISM metric; due to character constraints further details can be found in our **response to Reviewer 2jDP** under **[W5] Human evaluation**.
>
> **[Q4] Choice of the sentence embedding model:**
>
> We based the selection of the embedding model on the 'Massive Text Embedding Benchmark' (MTEB) that aims to evaluate sentence embedding models across a variety of tasks. Specifically, we chose a model from the Qwen2 family due to its strong and consistent performance on the MTEB leaderboard (see [MTEB23], [TEMB] for reviews, and [MTEB] for the current rankings).
>
> In initial experiments, we compared several top-performing embedding models and observed that the resulting concept descriptions showed strong overlap, and thus we chose not to pursue an extensive ablation in this direction. Overall, recent state-of-the-art encoder models produce embeddings with comparable performance [MTEB], reflecting a broader trend toward convergence in model representations that is hypothesized to be a results of strong similarities in training data, objectives, and model architectures [Plat24].
>
>
> **[Q5] Choice of LLM for description:**
>
> > "In practice, did you observe biases in your descriptions because of the choice of LLM (for summary generation) or sentence embedding function?"
>
> We agree that model choice can influence descriptions and is an important area for future research. In our early experiments, we observed that smaller LLMs often produced overly long or unfocused summaries, consistent with prior findings on their instruction-following limitations [Lou24, Zhe23]. For this reason, we use a strong and sufficiently large instruction-tuned model for summary generation. Due to character constraints, we kindly refer the reviewer to our **response to Reviewer MiB7** under **[W1/Q2] Open Source LLMs**, where we provide a detailed discussion and experimental results on the impact of using different LLMs for description generation.
>
> Regarding sentence embeddings, as discussed in [Q4], we found minimal variation in output semantics across high-performing embedder models. However, we acknowledge that domain-specific models (e.g., multilingual or biomedical) may be needed to capture sentence meaning in more specialized settings [Vas24].
>
>
> **[Q6] Distinction between mono- and polysemantic concepts:**
>
> We thank the reviewer for pointing out this potential for confusion. We follow the most widely used definition of polysemantic features as “neurons that respond to multiple unrelated inputs” [Fal15, Mul16, Aro18].
>
> In our manuscript, we thus consider a feature polysemantic if it activates for semantically distinct clusters (e.g., the mentioned encryption terms and holiday-related phrases), quantifiable  via the polysemanticity score. In contrast, features activating for different animal names are considered as monosemantic, since the underlying concept is coherent.
>
> We have clarified these points by (i) emphasizing and referencing the definition of polysemanticity used in our paper, (ii) expanding our discussion on the distinction between polysemantic and monosemantic features in Sections 5 and 6, (iii) extending our user study, which offers a complementary perspective on the semantic coherence of the identified feature clusters, and (iv) adding additional examples of clearly polysemantic and monosemantic features to the main paper.
>
>
> If any further questions or comments arise, we are happy to address them during the discussion period.
>
>
> **Refs**
> - [MTEB23] Muennighoff et al., MTEB: Massive Text Embedding Benchmark. EACL 2023.
> - [MTEB] Leaderboard accessed via huggingface.co/spaces/mteb/leaderboard
> - [TEMB]  H Cao, Recent advances in text embedding: A Comprehensive Review of Top-Performing Methods on the MTEB Benchmark. Arxiv 2024.
> - [Plat24] Huh et al., Position: The Platonic Representation Hypothesis. ICML 2024.
> - [Lou24] Lou et al., Large Language Model Instruction Following: A Survey of Progresses and Challenges. ACL 2024.
> - [Zhe23] Zheng et al., Judging llm-as-a-judge with mt-bench and chatbot arena. Neurips 2023.
> - [Vas24] Vasileiou et al., Explaining Text Similarity in Transformer Models. NAACL 24.
> - [Fal15]  Falkum  \& A Vicente., Polysemy: Current perspectives and approaches. Lingua 2015.
> - [Mul16] Nguyen et al., Multifaceted Feature Visualization: Uncovering the Different Types of Features Learned By Each Neuron in Deep Neural Networks. Visualization workshop ICML 2016.
> - [Aro18] Arora et al., Linear algebraic structure of
> word senses, with applications to polysemy. TACL 2018.

---

> > ### Comment · Reviewer_axts · 2025-08-02
> > **Rebuttal acknowledgement**
> >
> > Dear Authors,
> > Thank you for the rebuttal and clarifications. It addresses most of my concerns.
> >
> > I do however have one concern remaining regarding [W1/Q1]
> >
> > I understand and agree with the merits of your percentile sampling strategy over top-k strategies. However, my original point was **not** about absolute activations at all but rather **relative activations between 99th and 100 percentile samples for a given neuron**. To rewrite, assume for a given neuron $j$ in some layer and given dataset of samples, the 100th percentile activation is $a^j_{max}$ and 99th percentile activation is $a^j_{99}$. If $r_j = |\frac{a^j_{99}}{a^j_{max}}| \in [0,1]$ is very small (say like 0.05), then I don't think its a good strategy to consider 99th percentile sample as part of generating descriptions because it is not representative of a sample "highly activating" the neuron $j$.
> >
> >
> > Unless I am missing some fundamental point here for why the 99th percentile sample should be considered for a neuron even when $r_j$ is small, the best way to analyze if/how much this is a limitation would be to simply plot the distribution of $r_j$ for all the neurons (in a given layer/model for instance). The concrete question I ask is **what fraction of neurons have $r_j < \tau$ (say threshold $\tau=0.3$)?**

---

> > > ### Author Response · Authors · 2025-08-05
> > >
> > > We thank the reviewer for clarifying their question. In response, we have performed an analysis of $r_j$ across three layers (0, 20, and 40) in GPT-2 XL, using the same settings as in our main experiments. In Layers 0 and 20, none of the neurons exhibit $r_j < 0.3$. In Layer 40, we overall observe lower ratios $r_j$, but none fall below $0.05$. This experiment indicates that such cases are rare in practice, especially in early layers. In addition, in our evaluation across layers, we see that our method is able to robustly extract feature descriptions, even for more diverse ratios of $r_j$ (see also Figure 3 (a) in the main paper).
> > >
> > > To further illustrate that sampling from diverse activation ranges is practically meaningful, we provide additional examples. The table below shows an example where feature descriptions (neuron 3815, layer 47) in GPT-2 XL with diverse ranges of mean cluster activations (MeanAct), while high description scores (AUC) are maintained (see also Figure 1 of our main paper).
> > >
> > > | Description |MeanAct |AUC |
> > > |------------|-----------------|-----------------|
> > > |Quantities, specifically numbers, and time periods | 0.46 | 0.99 |
> > > |Personal experiences or opinions | 0.42 |0.98 |
> > > |Indefinite articles before contextual nouns |0.75 |0.99 |
> > >
> > > We add this additional example, alongside others (similar to the one above) in the Appendix of the revised manuscript.
> > >
> > > Based on the experiment and the examples, we argue that both theoretically and empirically, the percentile sampling reflects a focus on diverse functional and meaningful patterns, if present, beyond capturing isolated peak responses, as our goal is to retrieve multiple distinct patterns, resulting in multiple descriptions per feature.

---

> > > > ### Comment · Reviewer_axts · 2025-08-05
> > > >
> > > > Thank you for the reply
> > > >
> > > > It is indeed a really good sign you don't observe such neurons in early layers. Could you please also report what fraction of neurons had $r_j < 0.3$ for layer 40? I naturally expected earlier that at least some small fraction (in any layer) would possess this characteristic but I am trying to get some estimate about its range. Please do not worry that I am trying to attack the work by identifying some specific limitation. I will update my score accordingly taking into account the complete discussion and rebuttal.

---

> > > > > ### Author Response · Authors · 2025-08-05
> > > > >
> > > > > We thank the reviewer for their kind feedback! In Layer 40, we found that 75% of neurons we analyzed have $r_j < 0.3$.

---

> > > > > > ### Comment · Reviewer_axts · 2025-08-05
> > > > > >
> > > > > > Thank you to authors for the quick update.
> > > > > > All my questions from the review have been answered by them.

---

### Author Response · Authors · 2025-08-08

We thank the reviewers for their constructive feedback and active engagement during the discussion period. In response to their comments, we summarize the key changes below:

- **Ablations and complementary benchmarking**: In response to Reviewers axts, MiB7, and 9BmE, we added ablations using three additional open-source LLMs, with results discussed in Section 4.2.
Reviewers 9BmE and 2jDP were interested in comparisons to MaxAct that we now added to Table 1. Reviewer 9BmE  expressed interest in results on an output-centric faithfulness metric, which we now included in Table 1, Section 4.3. All results confirmed the robust performance and improved quality of our proposed multi-concept feature descriptions.

- We extended our **human evaluation study**  and its discussion in Section 5.2 as suggested by Reviewers axts, 2jDP and 9BmE. Results confirmed the agreement between user ratings and our proposed polysemanticity score.


- In response to Reviewers axts and 2jDP, we expanded on the **choice of the number of clusters** in Section 4.1 and now discuss the implications of varying $k$ in Section 6, referencing supporting experiments in the Appendix.


- We clarified **assumptions underlying our evaluation metrics** in Section 6 in response to Reviewer 9BmE.


- Based on Reviewer 9BmE’s comments, we **clarified the scope** of our work on text-based feature descriptions, with specific revisions made to the Abstract and Section 1.


Throughout the rebuttal, all reviewers recognized the novelty and practical relevance of our framework for addressing polysemanticity in LLMs. They found the method clearly presented (Reviewers axts, MiB7 and 9BmE), and the paper well-structured and readable (Reviewers  9BmE and axts). We appreciate Reviewer 9BmE’s remark that the "framework could easily become one popular baseline," and their assessment that our multi-level explanations are "an important feature of the framework and one of its strongest points". Our experimental evaluation was appreciated as "extensive" and "very thorough" (Reviewers axts and MiB7).

In summary, all requested ablations during the rebuttal (1) confirmed the robustness of our approach for generating and evaluating feature descriptions, (2) demonstrated the practical usefulness of our multi-concept description framework, and (3) aided in clarifying the scope of our work and communicating the utility and value of PRISM to the community.

---

### Note · Authors · 2025-08-11

We thank the reviewers for their constructive feedback and active engagement during the discussion period. In response to their comments, we summarize the key changes below:

- **Ablations and complementary benchmarking**: In response to Reviewers axts, MiB7, and 9BmE, we added ablations using three additional open-source LLMs, with results discussed in Section 4.2.
Reviewers 9BmE and 2jDP were interested in comparisons to MaxAct that we now added to Table 1. Reviewer 9BmE  expressed interest in results on an output-centric faithfulness metric, which we now included in Table 1, Section 4.3. All results confirmed the robust performance and improved quality of our proposed multi-concept feature descriptions.

- We extended our **human evaluation study**  and its discussion in Section 5.2 as suggested by Reviewers axts, 2jDP and 9BmE. Results confirmed the agreement between user ratings and our proposed polysemanticity score.


- In response to Reviewers axts and 2jDP, we expanded on the **choice of the number of clusters** in Section 4.1 and now discuss the implications of varying $k$ in Section 6, referencing supporting experiments in the Appendix.


- We clarified **assumptions underlying our evaluation metrics** in Section 6 in response to Reviewer 9BmE.


- Based on Reviewer 9BmE’s comments, we **clarified the scope** of our work on text-based feature descriptions, with specific revisions made to the Abstract and Section 1.


Throughout the rebuttal, all reviewers recognized the novelty and practical relevance of our framework for addressing polysemanticity in LLMs. They found the method clearly presented (Reviewers axts, MiB7 and 9BmE), and the paper well-structured and readable (Reviewers  9BmE and axts). We appreciate Reviewer 9BmE’s remark that the "framework could easily become one popular baseline," and their assessment that our multi-level explanations are "an important feature of the framework and one of its strongest points". Our experimental evaluation was appreciated as "extensive" and "very thorough" (Reviewers axts and MiB7).

In summary, all requested ablations during the rebuttal (1) confirmed the robustness of our approach for generating and evaluating feature descriptions, (2) demonstrated the practical usefulness of our multi-concept description framework, and (3) aided in clarifying the scope of our work and communicating the utility and value of PRISM to the community.

---

### Decision · Program_Chairs · 2025-09-17

**Decision:**

Accept (poster)

**Comment:**

The paper introduces PRISM, a framework to generate automated explanations for features or neurons within LLMs, with a specific focus on capturing polysemanticity. The method operates by employing percentile sampling to identify top activating samples for a given feature, followed by clustering these samples based on semantic similarity. A language model is then used to generate summaries for each cluster, effectively providing multiple explanations for a single feature. The framework is evaluated on various LLMs and demonstrated on neuron activations, with experimental results assessed using metrics like AUC, MAD, and a proposed Polysemanticity Score.

Reviewers generally appreciated the paper's clear writing and novel approach. The method's ability to address the interesting and relevant problem of capturing polysemanticity in LLM features was seen as a strength. Concerns were raised regarding the need for high-capacity language models for generating descriptions and the small size of the user study to measure alignment to human interpretation of polysemanticity. The authors addressed these concerns somewhat during the rebuttal period. Overall the reviewers were positive about the contributions of this paper in the field of automated interpretability. The authors are encouraged to revise the paper in light of reviewer feedback.